# Pneumolysin as a target for new therapies against pneumococcal infections: A systematic review

**María Dolores Cima Cabal[1], Felipe Molina[2], José Ignacio López-Sánchez[1], Efrén Pérez-Santín[1], María del Mar García-Suárez[1]***

**1** Escuela Superior de Ingeniería y Tecnología (ESIT), Universidad Internacional de La Rioja, UNIR, Logroño, La Rioja, Spain, **2** Genética, Facultad de Ciencias, Universidad de Extremadura, Badajoz, Spain

* mar.garcia.suarez@unir.net

## Abstract

### Background

This systematic review evaluates pneumolysin (PLY) as a target for new treatments against pneumococcal infections. Pneumolysin is one of the main virulence factors produced by all types of pneumococci. This toxin (53 kDa) is a highly conserved protein that binds to cholesterol in eukaryotic cells, forming pores that lead to cell destruction.

### Methods

The databases consulted were MEDLINE, Web of Science, and Scopus. Articles were independently screened by title, abstract, and full text by two researchers, and using consensus to resolve any disagreements that occurred. Articles in other languages different from English, patents, cases report, notes, chapter books and reviews were excluded. Searches were restricted to the years 2000 to 2021. Methodological quality was evaluated using OHAT framework.

### Results

Forty-one articles describing the effects of different molecules that inhibit PLY were reviewed. Briefly, the inhibitory molecules found were classified into three main groups: those exerting a direct effect by binding and/or blocking PLY, those acting indirectly by preventing its effects on host cells, and those whose mechanisms are unknown. Although many molecules are proposed as toxin blockers, only some of them, such as antibiotics, peptides, sterols, and statins, have the probability of being implemented as clinical treatment. In contrast, for other molecules, there are limited studies that demonstrate efficacy in animal models with sufficient reliability.

### Discussion

Most of the studies reviewed has a good level of confidence. However, one of the limitations of this systematic review is the lack of homogeneity of the studies, what prevented to carry out a statistical comparison of the results or meta-analysis.

**Data Availability Statement:** All relevant data are within the paper and its Supporting Information files.

**Funding:** This work was supported by Universidad Internacional de La Rioja under the project Pneumo-SARS UNIR-B0036 (2021–2022). The authors declare that they have no known competing financial interests or personal relationships that could have appeared to influence the work reported in this paper.

**Competing interests:** The authors declare no competing interest.

## Conclusion

A panel of molecules blocking PLY activity are associated with the improvement of the inflammatory process triggered by the pneumococcal infection. Some molecules have already been used in humans for other purposes, so they could be safe for use in patients with pneumococcal infections. These patients might benefit from a second line treatment during the initial stages of the infection preventing acute respiratory distress syndrome and invasive pneumococcal diseases. Additional research using the presented set of compounds might further improve the clinical management of these patients.

## Introduction

*Streptococcus pneumoniae* is the leading cause of community-acquired pneumonia in both adults and children [1]. It is also a common cause of meningitis and septicemia as well as other minor infections such as sinusitis and otitis media [2], even despite the extensive vaccination programs that exist today, especially in developed countries [3]. The prevalence of this bacterium is associated with its virulence factors and the patient's own risk factors including age, smoking, and other types of comorbidities (*e.g.*, diabetes or immunodeficiency). Virulence of *S. pneumoniae* is due to several factors, some of them related to the structure of the bacteria, such as the capsular polysaccharide (whose differences are related to bacteria serotypes) and the variety of surface protein families (*i.e.*, lipoproteins, sortase-anchored proteins, choline-binding proteins, and the non-classical surface proteins), but also the cytoplasmatic toxin, pneumolysin (PLY). The nature and action modes of these virulence factors, directly involved in the pathogenicity of pneumococcus, have been previously described elsewhere [1, 2, 4–6].

PLY is a toxin that binds to eukaryotic membrane cholesterol (belongs to cholesterol-dependent cytolysins, CDC) but also binds to the mannose receptor C type 1 (MRC-1) promoting an anti-inflammatory response and reducing pneumococcal disease [7, 8]. In this way, this toxin has double functionality ("sword and shield" or "Yin and Yang") [9]. PLY is produced constitutively but free toxin is higher in the late log phase due to the presence of a defined threshold concentration of extracellular autolysin (LytA) which dictates the onset of autolysis. The entry into the stationary phase due to nutrient depletion sensitizes cells to the effect of LytA, while during exponential growth they are protected from the action of this enzyme [10, 11]. Other investigations have also revealed the release of PLY in the extracellular vesicles [12]. On the other hand, a phenotypic heterogenicity has been demonstrated in terms of the level of expression of PLY, which helps the dispersion of the pneumococcus through the host [9, 13]. PLY does not have attachment motifs, however the toxin localizes to the cell envelope of actively growing cells, where its release and activity is controlled by the composition of the peptidoglycan, specifically by the proportion of branched stem peptides that vary throughout the cell cycle and between different strains [14].

PLY belongs to the family of thiol-activated toxins, commonly produced by many Gram-positive bacteria, which creates membrane pores in eukaryotic cells by binding to cholesterol, thus causing cell destruction [15]. It is worth noting that serotypes 1 and 8 have been shown to harbor mutations in the *ply* gene that annul this main characteristic and cause a much milder disease, due to a non-hemolytic allele (sequence type, ST306) allows adaptation to an intracellular lifestyle [16, 17]. However, in sub-Saharan Africa, serotype 1 causes invasive pneumococcal disease due to an increased production of autolysin and hemolytic pneumolysin alleles

[18]. (ST217). This serotype is a major cause of invasive pneumococcal disease globally, especially in Africa, South America, and Asia, with geographically distinct sequence types (STs) that form three genetic clusters designated as lineage A, B, and C [19].

Structurally, PLY consists of 471 amino acids (53 kDa) and has a three-dimensional conformation with 4 different domains; three of them (domains 1, 2, and 3) have structural importance, conferring stability to the PLY molecule, and are essential for oligomerization and pore formation in eukaryotic cell membranes. Domain 4 (hydrophobic region, loops L1-L3) forms the C-terminal region and is what promotes cholesterol binding, favoring the insertion of an undecapeptide sequence in the membrane, which allows for the oligomerization of toxin monomers, and the further formation of pores. The PLY pore is a 400 Å ring of 42 membrane-inserted monomers [20]. (**S1a Fig**) [21, 22].

This toxin can activate different cell death pathways such as apoptosis [23], pyroptosis [24], or necroptosis [25], which releases membrane-derived vesicles (microvesicles and exosomes). The pores of PLY cause a strong mitochondrial calcium influx which triggers mitochondrial morphological alterations with the release of mtDNA through microvesicles [26] and could regulate innate immune responses [27] (**S1a Fig**). During the inflammatory process, PLY activates several signal transduction pathways such as the nuclear factor kappa-B (NF-κB), mitogen-activated protein kinase (MAPK), and the NOD-like receptor pyrin domain-containing 3 (NLRP3) inflammsone [28, 29]. The interaction between PLY and TLR4 remains controversial. Some authors showed that the interaction of PLY with TLR4 is involved in induce cytokine production and apoptosis [30, 31], while other research showed that PLY activates the NLRP3/ACS inflammasome to enhance the secretion of pro-inflammatory cytokines IL-1β and IL-18 from macrophages and dendritic cells and contributes to the protection of the host from pneumococcal infection independent of TLR-4 and mediated by $K^+$ influx [24, 32].

Therefore, PLY is able to activate and regulate a huge number of genes for chemokines, cytokines, and other molecules whose expression is involved in the recruitment of inflammatory cells via neutrophils, macrophages, and phagocytes activation (such as IL-8, MCP-3, MIP-1β, lysozymes, caspases, TNF, IL-1 and IL-6) [33, 34]. Overproduction of early cytokines has been associated with tissue injury, organ dysfunction, morbidity, and mortality [35].

On the other hand, PLY is the main cause of pulmonary permeability edema due to its ability to alter both endothelial and epithelial barrier function [36]. This effect is a consequence of reducing dynamic and stable microtubule content in the endothelial monolayer and influencing VE-cadherin expression [37]. NLRP3 protect the alveolar barrier againts PLY injury [38]. Several cellular channels or transporters of the alveolar epithelium are implicated, such as the epithelial sodium channel (ENaC), the $Na^+$/ $K^+$ -ATPase and $K^+$ channels [39] (**S1a Fig**).

Moreover, PLY play a key role in human nasopharynx colonization and in the transmission of *S. pneumoniae* from host to host since by promoting inflammation there is an increase in elimination [40]. In addition, *S. pneumoniae* can invade other parts of the body via bloodstream dissemination, and it can gain access to normally 'sterile' sites such as the lower airways or meninges [41]. (**S1b Fig**). When bacteria reach the cerebrospinal fluid, produce meninges inflammation, resulting in hyperemia and ischemia, and eventually permanent brain injury [42].

Finally, it has been found that CAP (community acquired pneumonia) can incite up to 30% of the cases to cardiovascular events, such as myocardial damage or pro-thrombotic effects [43]. Myocardial damage is closely related to the ability of PLY to form pores in the cell membrane: movement of $Ca^{2+}$ into the cell leads to an efflux of $K^+$ with the consequent depolarization of the membrane, contributing to myocardial contractile dysfunction [44]. Regarding prothrombotic effects, this has recently been shown not to be true. PLY does not activate

platelets to form thrombus, rather it destroys them by forming pores in their membrane [45] and destroy procoagulant microvesicles impaired coagulation of blood [46].

Current pneumococcal conjugate vaccines are 13-valent conjugate vaccine (PCV13) for routine pediatric immunization and a 23-valent polysaccharide vaccine (PPSV23) for adults aged ≥65 years. Since 2014, PCV13 was also recommended for all adults aged ≥65 years [47]. In 2021, two new vaccines were approved by the FDA for use in adults ≥18 years (PCV-15 and PCV-20) that are presently under evaluation [48]. Although these vaccines are immunogenic and effective and prevent disease caused by the serotypes determined by their capsule types, they do not cover the full spectrum of invasive pneumococcal serotypes. The management of pneumococcal infections in clinical practice frequently involves the use of broad-spectrum antibiotics, typically a combined therapy of β-lactams and macrolides [49]. However, *S. pneumoniae* has developed resistance to multiple antibiotics including penicillin, macrolides, fluoroquinolone, and sulfamethoxazole-trimethoprim [50]. The emergence of non-vaccine serotypes after the introduction of PCV, together with increased antibiotic resistance in these serotypes, has become a global threat [51]. Taking together the development of new therapies is necessary for the effective treatment of pneumococcal infections.

Even though PLY-neutralization strategies have been reviewed before, these are partial reviews, that approach the subject from different perspectives, and the vast majority of these reviews focus on vaccination-based strategies (see for example [3, 52–56]). Furthermore, several previous reviews have been carried out on small-molecule-based PLY neutralization strategies that either suffer from being focused on a single family of compounds and do not represent complete overviews of the current state of the art, such as the work of Nishimoto et al. (2020) [57] which is focused on statins, the work of Li et al. (2017) [58], focused on sterols, or the work of Anderson and Feldman (2017) [59], which although it encompasses a broader collection of compounds, it cannot be considered complete and current to date.

The aim of this review was to investigate the main molecules that directly and indirectly interfere with PLY activity in host survival, colonization, infection, and transmission. A secondary objective was to identify the mechanisms of interaction with the toxin.

## Materials and methods

This systematic review was conducted in accordance with the Preferred Reporting Items for Systematic Reviews (PRISMA) guidelines [60] as much as possible, since it is not a medical systematic review, so part of the guidelines were not applicable (**S1 Table**).

### Search strategy and selection criteria

The inclusion criteria of the studies for the review were articles in which PLY was examined as a therapeutic target through studies of molecules that inhibit its effects. To analyze the therapeutic role of PLY in pneumococcal infection, the terms "pneumolysin" AND "therapeutic" were searched in three databases: Web of Science, MEDLINE and Scopus. At first, we only limited the search date to November 25, 2021; however, in the exclusion criteria, we decided to refine and not include articles prior to the year 2000. Therefore, the review includes the articles found from January 1st, 2000 to November 25th, 2021. Reference lists of selected articles were also reviewed. Several exclusion criteria were taken to ensure the quality of the study. The following were excluded: patents, articles in other languages different from English, cases report, notes, chapter books, reviews, and articles prior to 2000 year. Potentially eligible articles were reviewed by two independent reviewers (MMG-S and MDCC). Disagreements between the two reviewers were resolved by consensus.

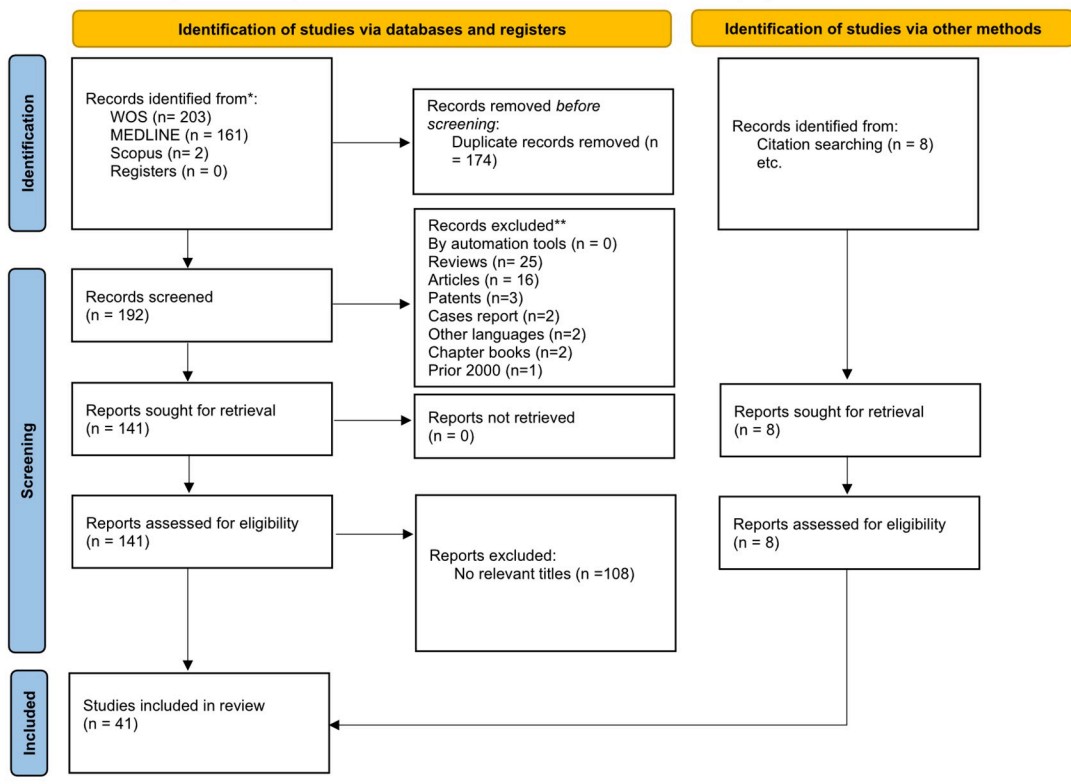

**Fig 1. Flowchart of the systematic review process.**

### Selection procedure

Citations were saved in the reference manager EndNote (https://endnote.com/) and in an Excel file that also included the following items: publication type, authors, article title, abstract, source title, DOI, and publication date. EndNote was used to deduplicate the search results. Full texts of these articles were analyzed by the two authors (MMG-S and MDCC). Articles on the pathogenesis of pneumococcus in general and articles in which PLY was presented as a vaccine and as tumor therapy were excluded from the study. **Fig 1** shows an overview of process used for the selection of the articles included in this review.

### Data collection process

Two authors collected data from articles in an electronic spreadsheet (**S2 Table**). The following data were obtained from each article: tests performed on cell lines and laboratory animals, CAS number and tested doses of the compound, route of inoculation, type of infection, and strain of *S. pneumoniae* used as well as other data considered of interest.

### Data analysis and synthesis

Quantitative and qualitative outcome data were provided and synthesized where possible. Due to data heterogeneity a meta-analysis was not possible.

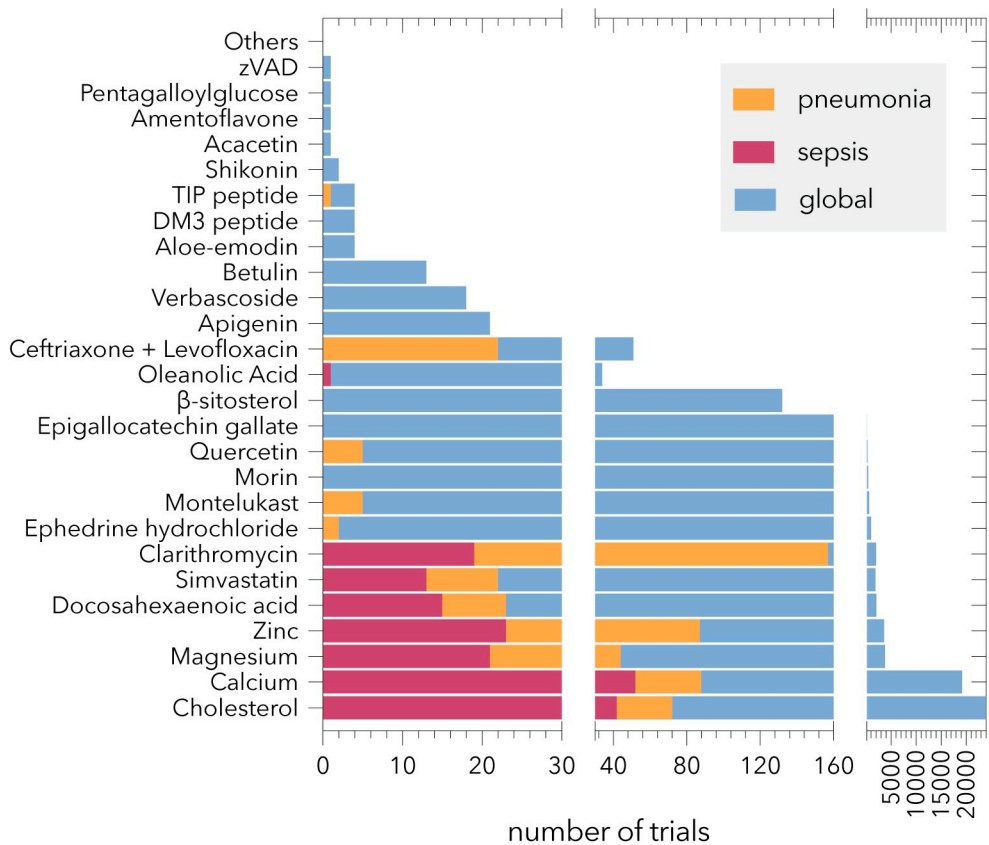

**Fig 2. Number of papers indexed in MEDLINE as clinical trials related to the molecules included in this review.** The molecules for which no clinical trials were found are grouped as "Others". The X-axis is split in three parts.

## Article quality assessment

Each article was scrutinized to determine the techniques that led to each molecule being proposed as a candidate for toxin-blocking therapy. The risk assessment tool used was the Office of Health Assessment & Translation (OHAT) risk of bias tool by the US Department of Health and Human Services [61]. Risk assessment was based on study type. This screening was tabulated in **S3 Table**. Moreover, we also inspected the number of papers of clinical trials indexed in MEDLINE for each proposed molecule (**S4 Table**; **Fig 2**).

## Network of neutralizing PLY-effects molecules and mechanisms underlying

The knowledge network harboring the molecules, the processes affected, and the cellular targets studied in this review were all generated using the tool EXTRACT 2.0 [62] to gather the terms from the articles' abstracts. From the resulting dataset, a graph was built using Cytoscape 3.9.1 [63], using the yFiles layout algorithms. Finally, some images were obtained from BioRender.com (2022).

## Results and discussion

### Study selection

A total of 366 records were found using the search criteria, of which 203 were found in WOS, 161 in MEDLINE and 2 in Scopus (**Fig 1**). Once the duplicates were eliminated, 192 articles

were reviewed by title and abstract by two independent authors (MG and MC). Several exclusion criteria were taken into account to ensure the review quality. As a result, 25 reviews, 16 articles, 3 patents, 2 cases report, 2 articles in other languages different than English, 2 chapter books, and 1 note prior to the year 2000 were excluded. The full texts of the resulting 141 articles were reviewed in detail. This refined search showed that 108 studies did not meet the criteria and were excluded (74 reports about the effect of PLY as a vaccine, 3 reports focused on tumor therapy, 31 reports studying general pathogenesis). References from these relevant articles were also screened. Forty-one (n = 41) studies met the inclusion criteria (describing molecules that inhibited the effects of PLY) were included in the systematic review.

## Study characteristics

A detailed analysis of the 41 resulting articles was performed, in which the effects of PLY were studied using molecules that block it. Most of the studies were published by scientists from Southeast Asia, principally China (n = 17), followed by Europe (n = 10), and the USA (n = 9) (S4 Table). In 15 of the articles, the experiments were carried out using only cell cultures, while in 26, animal models of pneumococcal infection were used (24 articles induced a pneumonia model, of which two also induced systemic infection) [64, 65], one model of meningitis [66], two of keratitis [67, 68], one of atherosclerosis [69] and one of nasopharyngeal colonization [70]. The most frequently used animal models were mice for pneumoniaand rabbits for keratitis. Meningitis was tested in rats and systemic infections were tested in Zebrafish embryos.

In 17 articles, the cellular cultures used were the human lung alveolar epithelial cell line A549, the human bronchial epithelial cell line HBE1 and H441 (n = 2), mouse leukemic monocyte-macrophage cell line RAW264.7 (n = 3), human umbilical vein endothelial cells (n = 3), human lung microvascular endothelial cells (n = 2), neutrophils (n = 2), cochlear hair cells (n = 1), primary glial cultures (n = 1), and primary astrocytes (n = 1). Laboratory animals used to induce pneumococcal pneumonia were C57BL/6 mice (n = 12), BALB/c mice (n = 8), MF-1 mice (n = 1), ICR mice (n = 1), and C3H mice (n = 1).

## Methodological quality

The OHAT quality scores of the included articles are described in S3 Table. The overall risk of bias rating was considered as Low or Probably Low. The 41 studies were summarized and classified into 10 groups according to chemical structures: Plant-derived compounds, sterols, statins, omega-3 fatty acids, purin-6-ones, thioethers, antibiotics, peptides, cations, and antibodies (Table 1). Moreover, because of the high heterogeneity of the data found in the articles, it was very difficult to compare the action mechanism against the toxin from each study, so they were grouped into three categories for clarification: direct (the molecule binds to or competes with PLY for binding to the target), indirect (the molecule does not bind PLY), and unknown (molecular mechanisms are currently unknown).

On the other hand, Fig 2 shows the number of articles indexed as 'clinical trials' on MEDLINE (without time restrictions), illustrating that only 17 of 39 molecules present experimental human evidence to potentially conclude data on their efficacy, toxicity, or bioavailability. The distribution of studies regarding molecules is unequal. For some molecules, there are more than 1000 studies (like cases of calcium and cholesterol), for others, the number ranges from 50–100, and for 10 of them, there are less than 20 studies. As expected, more trials and global studies are found to specifically study sepsis and pneumonia molecules. Thus, most trials focused on sepsis are limited to molecules, exhibiting more than 1000 global studies.

**Table 1. Basic characteristics of the articles included into this report show the molecules with effects against PLY.**

| Reference | Molecule (No CAS) | Molecule type | Toxicity assays | Action mechanism | Infection type | Cellular lines and laboratory animals | *S. pneumoniae* strain |
|---|---|---|---|---|---|---|---|
| [77] | Acacetin (480-44-4) | Flavonoid, polyphenol (natural) | **Cytotoxicity assays:** Purified PLY (0.4 μM) was incubated with different concentrations of acacetin, 8, 16, 32 μg/ml, all of them significantly decreased the LDH release of A549 cells.<br>**Animal models:** Intranasal infected mice were injected subcutaneously with acacetin at 50 mg/kg. The levels of IL-1β and IFN-γ in bronchoalveolar lavage fluid and the wet/dry weight ratio of lung tissue was significantly decreased at 96 h post-infection. | **Direct.** Acacetin inhibit the oligomerization of PLY. Iteration residues no shown. | Pneumonia | BALB/c mice | D39 |
| [78] | Amentoflavone (1617-53-4) | Flavonoid, polyphenol (natural) | **Cytotoxicity assays:** A549 cells were treated with PLY (0.825 μg/ml) and increasing concentrations of amentoflavone (AMF). LDH release significantly decreased at AMF concentrations higher than 2 μg/ml.<br>**Animal models:** Mice were administered with AMF (50 mg/kg) subcutaneously 2 h after infection with D39. Analysis of lungs 72 h after infection showed significant improvement of inflammatory cell accumulations and alveolar damage with respect to the control mice. | **Direct.** Amentoflavone interacts with the toxin at Ser254, Glu277, Arg359, and weakens the oligomerization of PLY. | Pneumonia | A549 cells[1] C57BL/6J mice | D39 |
| [79] | Morin (480-16-0) | Flavonoid, polyphenol (natural) | **Cytotoxicity assays:** Morin reduces the cytotoxicity of PLY (0.2 mg/ml) in cells at concentrations above 2 μg/ml.<br>**Animal models:** Infected mice were injected subcutaneously with 50 mg/kg morin two hours after infection. Three days post-infection, the lung tissues in the morin treatment group of mice displayed a significant reduction of inflammatory cell infiltration, bacteria number, and pulmonary inflammation. | **Direct.** Morin inhibits the oligomerization of PLY. Iteration residues no shown. | Pneumonia | A549 cells C57BL/6J mice | D39 |
| [76] | Apigenin (520-36-5) | Flavonoid, polyphenol (natural) | **Cytotoxicity assays:** A549 cells were exposed to 80 nM PLY and different concentrations of apigenin (0, 10, 20, 40 and 80 μM). Apigenin at a concentration range of 40 to 80 μM conferred significant protection against toxin cellular effects.<br>**Animal models:** Mice were intranasally infected and 2 h later subcutaneously administered apigenin (80 mg/kg). At 48 h post-infection, the bacterial burden, the tumor necrosis factor α (TNF-α) and interleukin 1β (IL-1β) in the bronchoalveolar lavage fluid levels were significantly reduced compared with the levels detected in the control mice. | **Direct.** Apigenin Inhibits the oligomerization of PLY. Iteration residues no shown. | Pneumonia | A549 cells BALB/c mice | D39 |
| [80] | Epigallocatechin gallate (989-51-5) | Flavonoid, Polyphenol (natural) | **Cytotoxicity assays:** A549 cells were pre-incubated with various concentrations of Epigallocatechin gallate (EGCG) (0, 1.09, 2.18, 4.36, and 8.73 μM) and incubated with PLY (80 nM). Concentrations above 1.09 μM showed significant differences in LDH release.<br>**Animal models:** Mice were intranasally inoculated to produce pneumonia and were subcutaneously administered EGCG (50 mg/kg). The mortality at 120 h decreased approximately a 40%. | **Direct.** Epigallocatechin gallate inhibits the oligomerization of PLY. The binding model of EGCG with PLY revealed that the side chain of EGCG can form strong interactions with Ser256, Glu277, Tyr358 and Arg359. | Pneumonia | A549 cells Hep2[2] BALB/c mice | D39 |
| [81] | Quercetin (117-39-5) | Flavonoid | **Cytotoxicity assays:** A549 cells incubated with 3 μl PLY (0.16 mg/ml) and preincubated with Quercetin (2, 4, 8, 16, 32 μg/ml) significantly reduced PLY cytotoxicity at all concentrations assayed.<br>**Animal models:** Infected mice were subcutaneously treated with Quercetin (25 mg/kg). After 96h post-infection, the survival rate of treated mice was 80% versus 60% in control mice. Edema and cytokine release were decreased. | **Direct.** Quercetin inhibits PLY oligomerization. Iteration residues no shown. | Pneumonia | A549 cells BALB/c | D39 |
| [82] | Dryocrassin ABBA (12777-70-7) | Flavonoid | **Hemolysis:** PLY (0.4 μM) was incubated with ABBA (0, 1, 2, 4, and 8 μg/ml). Reduction of hemolytic activity was observed at concentrations above 2 μg/ml.<br>**Cytotoxicity assays:** Cells incubated with ABBA (2, 4, 8, 16 μg/ml) and PLY (0.4 μM) reduced PLY toxicity at 16 μg/ml. | **Direct.** Quercetin inhibits PLY oligomerization. Iteration residues no shown. | | RAW264.7 cells[3] | D39 |
| [75] | Verbascoside (61276-17-3) | Phenylpropanoid glycoside, Polyphenol (natural) | **Cytotoxicity assays:** LDH release assay was performed to evaluate the effect of Verbascoside (VBS) on PLY-mediated lysis of A549 cells. The addition of 2 to 32 μg/ml of VB) reduced the cytotoxicity of PLY in a dose-dependent manner.<br>**Animal models:** Mice infected with *S. pneumoniae* were treated with 100 mg/kg VBS. Survival rate at 120 h after infection was 75% comparing with the 20% untreated control group. | **Direct.** Verbascoside inhibits the oligomerization of PLY binds to the cleft between domains 3 and 4 of PLY.<br>The hydroxyl group of the benzene ring on the right side of VBS can form a strong hydrogen bond with Asp471 which has a crucial significance in making the right side of VBS stable. | Pneumonia | A549 cells C57BL/6J mice | D39 |

*(Continued)*

**Table 1.** (Continued)

| Reference | Molecule (No CAS) | Molecule type | Toxicity assays | Action mechanism | Infection type | Cellular lines and laboratory animals | *S. pneumoniae* strain |
|---|---|---|---|---|---|---|---|
| [83] | Pentagalloylglucose (14937-32-7) and gemin A (82220-61-9) $C_{82}H_{56}O_{52}$ | Tannins, polyphenols (natural) | **Hemolysis:** Inhibitory activity of PLY was calculated. The Pentagalloylglucose (PGG) $IC_{50}$ was $18 \pm 0.7$ nM and gemin A $IC_{50}$ was $41 \pm 1$ nM. **Cytotoxicity assays:** A549 cells were incubated with 2 nM of PLY and 500, 1000 and 2000 nM of PGG inhibited LDH release by 60%, 87% and 90%, respectively. | **Direct.** 27 hydrolysable tannins were tested. PGG and gemin A were the most active monomer and oligomer, respectively. PLY oligomerization on the erythrocyte surface was inhibited with PGG. Flexible galloyl groups of PGG binds to the pocket formed by domains 2, 3, and 4 of PLY (Glu42, Ser256, Asp257, Glu277 and Arg359). | | A549 cells | |
| [85] | Juglone (481-39-0) | Naphthoquinone | **Hemolysis:** 4nM PLY was mixed with Juglone (JG) (0, 2.6, 5.2, 10.4 and 20.8 µg/ml). Hemolytic activity was significantly reduced at 10.4 µg/ml. **Cytotoxicity assays:** Cells were incubated with 80 nM PLY and JG (0, 2.6, 5.2, 10.4 and 20.8 µg/ml). Significant reduction of cytotoxicity was observed at 10.4 µg/ml. | **Direct.** Juglone inhibits the oligomerization of PLY. Iteration residues no shown. | | A549 cells | D39 |
| [84] | Shikonin (517-89-5) | Naphthoquinone (natural) | **Cytotoxicity assays:** The A549 cells were combined with different concentrations of shikonin (0.5–16 µg/ml and PLY (0.2 µg). All concentrations of shikonin inhibited the toxic effect of PLY. **Animal models:** Mice with endonasal pulmonary infection were orally treated with shikonin (50 mg/kg). 5 days after infection, the percentage of survival in treated mice was 65% versus 10% in the control mice. | **Direct.** Shikonin inhibits the oligomerization of PLY. Shikonin could bind to PLY through contact with certain amino acid sites thus affecting the conformational transition for PLY from the monomeric to oligomeric form. | Pneumonia | A549 cells C57BL/6J mice | D39 |
| [71] | Aloe-emodin (481-72-1) | Anthraquinone (natural) | **Hemolysis:** *S. pneumoniae* cultures were treated with light alone (72 J/cm2) (P-L+), Aloe-emodin (AE) (32 µg/ml) alone (P+L-) or AE combined with light (P+L+) and supernatants added to erythrocytes. The hemolytic activity diminished 93.67% (P-L-) 83.33% (P-L+) 63.67% (P+L-) and 23.66% (P+L+). | **Unknown.** AE+ photodynamic therapy (435 ± 10 nm) reduced bacterial survival, biofilm formation, cytokine production, and inhibits PLY expression. | | RAW264.7 cells | ATCC 49619 MDR (ATCC 49619 |
| [86] | Ephedrine hydrochloride (50-98-6) Pseudoephedrine hydrochloride (345-78-8) | Phenethylamines (natural) | **Cytotoxicity assays:** A549 cells were cocultured with Ephedrine hydrochloride ESG (0, 4, 8, 16 and 32 µg/ml) preincubated with PLY (0.2 µg). LDH released was significantly reduced at 4 µg/ml. **Animal models:** In vivo protection was tested in infected mice at 72h. In mice subcutaneously treated with ESG (40 mg/kg), the percentage of survival was 50% versus 0% in the untreated control group. | **Direct.** ESG inhibits the oligomerization of PLY. Iteration residues no shown. | Pneumonia | A549 cells BALB/c mice | D39 |
| [87] | Ephedrine hydrochloride (50-98-6) Pseudoephedrine hydrochloride (345-78-8) Methylephedrine (552-79-4) Amygdalin (29883-15-6) Prunasin (99-18-3) Glycyrrhetinic acid (471-53-4) | Phenethylamines (natural) Amygdalin and Glycyrrhetinic acid cyanogenic glycoside Glycyrrhetinic acid steroid hormone | **Cytotoxicity assays:** A549 cells were pre-incubated with various concentrations of MXSGT (0, 4, 8, 16 or 32 µg/ml) and incubated with PLY (0.2 µg/ml). Concentrations above 8 µg/ml showed significant differences in LDH release compared with the control. **Animal models:** Percentage of survival of mice infected and treated with MXSGT (50 mg/kg) was 40% protection at 72h versus 0% in the untreated mice control group. | **Direct.** Inhibits the oligomerization of PLY. Iteration residues no shown. | Pneumonia | A549 cells BALB/c mice | D39 |
| [91] | Hederagenin (465-99-6) | Triterpenoid | **Hemolysis:** Hederagenin (2–32 µg/ml) was incubated with PLY (0.037 mg/ml). Hederagenin significantly inhibited the hemolytic activity of PLY at 8 µg/ml. **Cytotoxicity assays:** The protector effect was observed at concentrations above 16 µg/ml in cells incubated with hederagenin (0, 2, 8 and 32 µg/ml) and PLY. | **Direct.** Hederagenin interfered with the oligomerization of PLY. Iteration residues no shown. | | A549 cells RAW264.7 cells | D39 |

(*Continued*)

**Table 1.** (Continued)

| Reference | Molecule (No CAS) | Molecule type | Toxicity assays | Action mechanism | Infection type | Cellular lines and laboratory animals | *S. pneumoniae* strain |
|---|---|---|---|---|---|---|---|
| [90] | Betulin (473-98-3) | Triterpene | **Hemolysis:** 2 µl of PLY (0.1 mg/ml) was incubated with Betulin (0,1, 2, 4, 8 µg/ml). Inhibition of hemolysis was observed at concentrations above 2 µg/ml. **Cytotoxicity assays:** A549 cells were exposed to PLY (80 nmol/l) and Betulin (4, 8, 16, 32 µg/ml). Protection of cytotoxicity was dose dependent. DNA-damage was also protected above 8 µg/ml. | **Direct.** Betulin inhibited the oligomerization of PLY. Iteration residues no shown. | | A549 cells | D39 |
| [89] | Oleanolic Acid (508-02-1) | Pentacyclic triterpenoids (natural) | **Hemolysis:** PLY was mixed with different concentrations of Oleanolic Acid (OA) (from 0 to 32 µg/ml). Calculated $IC_{50}$ was 2.62 µg/ml. **Cytotoxicity assays and animal models:** This experimentation was only carried out for *S. aureus* and *E. coli*. | **Direct.** OA inhibits the action hemolytic of PLY. | Pneumonia | A549 cells C57BL/6J mice | |
| [67] | Cholesterol (57-88-5) | Sterol (natural) | **Hemolysis:** PLY at a concentration of 500 ng, 100 ng, 50 ng, 10 ng, 5 ng, or 0 ng was mixed with 1% cholesterol. Cholesterol at 1% inhibited 50 ng PLY by 92%. **Animal models:** Pneumococcal keratitis was induced in rabbit corneas by intrastromal injection of *S. pneumoniae* or recombinant PLY (1 µg). 25 h after infection, rabbit eyes were topically treated with a drop of 1% soluble cholesterol (40 mg). Seven ocular parameters were classified. Significant differences between cholesterol-treated corneas and untreated corneas were observed 48h post infection. | **Direct.** Cholesterol inhibits the PLY hemolysis. The hydrophobic region of domain 4 is responsible for interacting with cholesterol, specifically through a THR-LEU pair located in the L1 loop. | Keratitis | New Zealand White rabbits | D39 |
| [93] | β-sitosterol (83-46-5) | Sterol (natural) | **Cytotoxicity assays:** The protective effects of β-sitosterol (BSS) on the toxicity of A549 cells were measured by the release of LDH showing that concentrations higher than 2 µg/ml protect the injury caused by 20 µg of PLY. **Animal models:** Mice infected were treated 1 h prior to the infection with BSS (80 mg/kg); a 70% survival rate at 5 days was observed. | **Direct.** β-sitosterol interacts with the toxin at THR459 and LEU460, competing with cellular cholesterol for binding to the toxin. | Pneumonia | A549 cells C57BL/6 mice | D39 |
| [97] | Simvastatin (79902-63-9) | Statins (Synthetic) | **Cytotoxicity assay:** 1 µM simvastatin provided significant protection against higher doses of PLY, ranging from 0.4 to 1.6 µg/ml. 10 µM hydrophilic pravastatin also conferred significant protection against PLY. **Animal models:** 200 ng PLY was injected intratracheally in mice that have been injected intraperitoneally with simvastatin (20 mg/kg). Lungs of mice sacrificed 18 h later demonstrated lower injury and inflammation compared with the control group. | **Indirect.** Simvastatin significantly reduced the total cholesterol content of cells, but did not reduce the binding of PLY to cells. Simvastatin inhibits the cytotoxicity of PLY on airway epithelial cells. | Pneumonia | HBE1[4], NHBE[5], A549 cells C57BL/6 mice | |
| [99] | Simvastatin (79902-63-9) | Statins (Synthetic) | **Cytotoxicity assay:** Cells were preincubated overnight with 0, 0.1, 1, or 10 µM simvastatin and incubated with PLY 3 µg/ml. The protection effect was observed at 1 µM. **Animal models:** WT and sickle cell disease (SCD) mice were treated with simvastatin (1 µg/g via intraperitoneal) daily for 5 days prior to bacterial challenge. Simvastatin had no significant effect on the survival of infected WT mice whereas the time of death was significantly delayed in SCD-treated mice. | **Indirect.** Statin treatment reduced platelet-activating factor receptor (PAFr) expression. PAFr binds to bacteria and is engulfed in vacuoles, invading the host cell. | Pneumonia and sepsis | HBMEC[6] C57BL/J6 mice | D39 |
| [109] | Docosahexaenoic acid (6217-54-5) | Fatty acids Omega3 (natural) | **Hemolytic activity:** Percentages of hemolysis observed with PLY at 8.37 ng/ml alone or in the presence of 5 mg/ml Docosahexaenoic acid (DHA) were 56.3 ± 2% and 44 ± 1.4%, respectively. For PLY at 4.19 ng/ml in the presence of the same DHA concentration, the percentages of hemolysis were 33.3 ± 1.7% and 22.7 ± 1.3%, respectively. | **Direct.** DHA interferes with the binding of PLY to target cells. PLY-mediated influx of Ca2+, activation of NFκB (nuclear factor kappa-light-chain-enhancer of activated B cells) and IL-8 in neutrophils. | | Human neutrophils | |
| [116] | 9-(6-phenyl-2-oxohex-3-yl)-2-(3,4-dimethoxybenzyl)-purin-6one (190666-14-9) | Purin-6-ones (Synthetic) | **Transcellular electrical resistance (TER):** HUVEC monolayers were incubated with PDP (1 and 0.01 µM) followed by toxin exposure. PLY (0.1 µg) evoked a strong decrease of TER, indicating increased endothelial cell permeability. PDP preincubation concentration dependently reduced the PLY-evoked increase in permeability. **Animal models:** Intranasal pneumococci-infected mice were treated 1 hour before infection with hydroxy-PDP (61.93 ±11.77 nM concentration final) infused subcutaneously. 48h after infection, lung hyperpermeability was decreased; however, inflammatory cell number did not change. | **Indirect.** Phosphodiesterase 2 (PDE2) inhibition (with PDP or hydroxy-PDP) decreased PLY induced human endothelial cell and alveolo-capillary membrane permeability. Protein expression of PDE2 in lung homogenates of pneumococci-infected lungs was significantly increased 48 h after infection. PDE2 inhibition decreased lung hyperpermeability induced by PLY. | Pneumonia | HUVEC[7] C57Bl/6 mice | ST3 |

*(Continued)*

**Table 1.** (Continued)

| Reference | Molecule (No CAS) | Molecule type | Toxicity assays | Action mechanism | Infection type | Cellular lines and laboratory animals | S. pneumoniae strain |
|---|---|---|---|---|---|---|---|
| [118] | CysLT1 antagonists Montelukast (158966-92-8) | Drug (Synthetic) | **Animal models:** Among BLT2-knockout mice intratracheally injected with 50 ng of PLY, 80% survived compared with the control group. Montelukast (5 mg/kg) improves PLY-induced (50 ng) acute lung injury in mice. | **Indirect.** BLT2 is a G protein-coupled receptor for leukotriene B4 and 12-HHT is a natural ligand that protects mice from lung injury caused by PLY. BLT2 suppress CysLT1 (receptor of cysLTs) expression. PLY triggers the production of cysteinyl leukotrienes (cysLTs) that activate CysLT1 expressed in vascular endothelial cells and bronchial smooth muscle cells, leading to lethal vascular leakage and bronchoconstriction. Montelukast is a CysLT1-selective antagonist that reverses the effects of PLY: vascular leakage and bronchoconstriction. | Pneumonia | BALB/c mice | |
| [121] | Clarithromycin (81103-11-9) | Antibiotics– Macrolide (Synthetic) | **Animal models:** Mice were intratracheally infected with S. pneumoniae. Clarithromycin (150 mg/kg) and Erythromycin (150 mg/kg) were administered orally 150 mg/kg every 12 h. 24h post infection, arterial oxygen saturation was improved whereas neutrophil number, IL-6 and PLY in the BALF decreased as compared to the control. | **Unknown.** Clarithromycin significantly downregulates *ply* gene transcription compared with ERY and the control. | Pneumonia | BALB/c mice | NU4471 Macrolide resistant D39 |
| [122] | Ceftriaxone (104376-79-6) Levofloxacin (100986-85-4) | Antibiotics– Cephalosporin (Synthetic) Antibiotics– Quinolone (Synthetic) | **Animal models:** Mice were infected by intranasal challenge with multidrug-resistant S. pneumoniae and treated with intravenous doses of either Levofloxacin (LVX) (150 mg/kg) or Ceftriaxone (CRO) (50 mg/kg of body weight) or administered together. The percentage of survival at 72h was: LVX, 55%; CRO, 38%; and LVX+CRO 95% survival. | **Unknown.** Downregulating the expression of *ply* gene. | Pneumonia | BALB/c mice | Clinical isolate |
| [69] | C-terminal 70 amino acids of PLY (C70PLY) | Peptide | **Cytotoxicity assays**: HUVECs were treated for 24 h with PLY and C70PLY peptide at concentrations of 1, 10, 100, or 1000 nM. For PLY, cell viability was strongly diminished to 100 and 1000 nM (more than 40%) whereas no cell toxicity was observed for any C70 peptide concentration. **Animal models:** The anti-inflammatory potential of C70PLY peptide was determined in a model of atherosclerosis-induced rats. Rats treated with C70PLY reduced the formation of neointima which developed into atherosclerotic plaque in the control group. | **Direct.** PLY is a toll-like receptor (TLR) 4 ligand. TLR4 forms a complex with its specific coreceptor myeloid differentiation factor 2 (MD2). The peptide binds M2 inhibit toxin-induced lysis of host cells. | Atherosclerosis | HUVEC and PMN[8] Sprague–Dawley rats | |
| [64] | DM3 peptide: GLFDIWKWWRWRR-NH2 Indolicidin peptide derivative: ILAWKWAWWAWRR-NH2 | Synthetic hybrid peptides | **Animal models:** Two models were assayed: systemic and pneumonia pneumococcal infection. Several combinations of peptide and penicillin were tested. In the systemic infection model, the best combination obtained (100% survival) were observed in mice treated with DM3 (20mg/kg) + penicillin (20mg/kg) treatment. In the pneumonia infection model mice survival was not observed in any treatment. | **Direct.** DM3 has a strong affinity PLY (interaction with ALA370, TYR371, TYR376, ASN400, ASP403, CYS428, ALA432, TRP435, TRP436 of PLY) and showed therapeutic synergism in combination with penicillin. Indolicidin peptide derivative bounded better than DM3 (In silico). | Systemic and pneumonia | ICR mice | Penicillin Resistant strain from clinical isolate |
| [7] | Mannose receptor peptides (MRC-1) | Peptide | **Cytotoxicity assays:** Human THP-1 macrophages treated with PLY (0.5 μg/ml) and 100 μM peptides. Significant reduction of toxicity was observed (~ 50%) with P2 and P3. **Animal models:** Two models were assayed: mice pneumonia infection and zebrafish fertilized embryo infection. Mice were infected intranasally with bacteria mixed with peptide P2 alone (5 μg) or P2 conjugated to CaP NPs (5 μg peptide; 25 μg CaP NPs) (biocompatible calcium phosphate (CaP) nanoparticles (NPs) as peptide nanocarriers). Percentage of survival at 3 days were increased to 50% in mice treated with P2-NP and 20% at 2 days in mice treated with P2. Infected zebrafish fertilized embryos were microinjected into the yolk sac with 500 CFU mixed with P2 or P2-conjugated NPs (1nM). After 100h post-infection, only 50% of embryos survived the infection versus 80% of embryos treated with P2 or P2-NP. | **Direct.** The cholesterol binding loop of PLY (W433, W435, W436, E434) bind the C-type lectin domain 4 (CTLD4) of MRC-1. Peptides competitively inhibit the PLY binding. | Pneumonia and systemic | Human THP-1 macrophages C57BL/6 mice Zebrafish infection model | TIGR4 D39 |

*(Continued)*

**Table 1.** (Continued)

| Reference | Molecule (No CAS) | Molecule type | Toxicity assays | Action mechanism | Infection type | Cellular lines and laboratory animals | *S. pneumoniae* strain |
|---|---|---|---|---|---|---|---|
| [37] | TNF-derived tonoplast intrinsic protein (TIP peptide). PKC-α inhibitor Ro 32–0432 hydrochloride (Bisindolylmaleimide XI hydrochloride) (145333-02-4) | Peptides and drug (Synthetic) | **Measurement of Transendothelial Electrical Resistance (TER):** PLY treatment of human lung microvascular endothelial cells induces a dose-dependent increase in intracellular Ca2+ concentrations. Pretreatment of the cells for 1 hour with PKC-α inhibitor Ro32-4032 (10 nM) or with TIP peptide (27 μM) significantly reduces this effect. **Animal models:** Intratracheal instillation of 3.125 μg PLY/kg induces a significantly increased pulmonary endothelial permeability. Cotreatment of the mice with the PKC-α inhibitor Ro32-0432 (49 μg/kg, intratracheally) or with the TIP peptide (2.5 mg/kg, intratracheally) significantly inhibits PLY-induced capillary leak. | **Indirect.** PLY causes increased arginase activity and PKC-α activation, both involved in endothelial dysfunction. A specific PKC-α inhibitor (Ro 32–0432) and the TIP peptide inhibit PKC-α and arginase which respectively mitigates the endothelial hyperpermeability. | Pneumonia | HLMVEC[9] and HPAEC[10] C57BL/6 mice | |
| [129] | GHRH agonist: JI-34 peptide | Peptide analogs | **Measurement of Transendothelial Electrical Resistance (TER):** PLY treatment of H441 cells (30 ng/mL) leads to a significant reduction of both inward and outward currents. The subsequent treatment of the cells with JI-34 (1 μM) restores the Na+ current. **Animal models:** JI-34 (100 μg/kg) completely inhibits the increase in lung wet-to-dry weight ratio (edema) caused by PLY instillation (6.125 μg/kg). | **Indirect.** JI-34 can restore Na+ uptake and blunts phosphorylation of MLC (myosin light chain) and VE-cadherin induced by PLY. It also mitigates the endothelial hyperpermeability. | Pneumonia | H441 cells[11] and HLMVEC C57BL6 mice | |
| [130] | Vasculotide | Peptide modified | **Measurement of Transendothelial Electrical Resistance (TER):** Cells were pretreated with Vasculotide (VT) (2, 10, and 50 ng/ml) and then with PLY (0.75 μg/ml). Preincubation with VT at 50 ng/ml attenuated the PLY-induced TER. **Animal models:** Mice were transnasally inoculated with bacteria a nd i.v. with VT (100, 200, or 500 ng). VT 500 ng significantly reduced edema formation but leukocyte recruitment and cytokine production 48h post-infection were not affected. | **Indirect.** Angiopoietins are regulators of inflammation and vascular leakage and ligands for the receptor tyrosine kinase Tie2. VT bind Tie2 mimic Angiopoietin-1. | Pneumonia | Murine lung endothelial cells (mLEC) C57BL/6 N | NCTC7978 |
| [133] | Z-VAL-ALA-OH (24787-89-1) | Peptide modified | **Cytotoxicity assays**: HUVEC were pre-incubated with 50 μg/ml zVAD for 30 minutes and then stimulated with *S. pneumoniae* R6x. zVAD completely inhibited pneumococci-related DNA fragmentation (apoptosis). Pneumolysin-deficient mutant R6xΔply induced apoptosis less effectively than R6x. | **Indirect.** Deletion of the gene coding for PLY reduced pneumococci-induced apoptosis in HUVEC, involving mitochondrial death pathways. Programmed cell death could be strongly reduced by pan-caspase inhibitor zVAD. | | HPMEC-ST1.6R[12] and HUVEC | R6x |
| [31] | zVAD-fmk (187389-52-2) | Peptide modified | **Animal models:** Mice were intranasally inoculated with strain WU2 and zVAD-fmk (20 μg) every 12h. Percentage of survival was 50% in treated mice versus 90% in the control. This treatment diminished survival rate. | **Indirect.** PLY is a ligand for the TLR-4 receptor, activating the innate immune response. TLR-induced apoptosis is caspase 3 dependent and is a defense mechanism against infection. Thus, inhibition with zVAD-fmk decreases survival to infection. | Sepsis | C3H/HeOuJ and C3H/HeJ mice | WU2 and A66.1 Xen 10 |
| [66] | Mg2+ | Cation | **Cytotoxicity assay:** Mouse glial cells, 4 HU/ml PLY and 2 mM Mg treatment, diminished LDH release at 60 min after PLY challenge. **Animal models:** Two models of induced meningitis were assayed in mice and infant rats. Infant SD rats were intracerebroventricular (i.c.v.) injected with PLY (4 HU/ml) and i.p. with MgCl2 (500 mg/kg) which significantly reduced brain edema. Mice received an intracerebral (i.c.) injection of 1000 CFU and with three i.p. doses MgCl2 30.45 mg/ml prolonged survival at 36 h; mortality was 2/18 in the treated group versus 8/19 in the control group. | **Unknown**. Magnesium diminishes interstitial brain edema caused by PLY. Mg2+ diminishes the pore-forming capacity of PLY without inhibit toxin binding to cells. | Meningitis | Primary glial cultures C57BL/6 mice SD rats | D39 |
| [135] | Zn2+ | Cation | **Cytotoxicity assay:** Cells were incubated with PLY 1ng/μl. The protective effect of zinc at 1 μM was 39.3% and 50.22% for IHCs and OHCs, respectively. At 300 μM of zinc, protection was significantly increased with 62% and 55.2% for IHCs and OHCs, respectively. | **Unknown**. Zinc inhibits the incorporation of PLY into the membrane. | | Cochlear hair cells (HCs) and Inner hair cells (IHCs) Wistar rats | |

(*Continued*)

**Table 1.** (Continued)

| Reference | Molecule (No CAS) | Molecule type | Toxicity assays | Action mechanism | Infection type | Cellular lines and laboratory animals | *S. pneumoniae* strain |
|---|---|---|---|---|---|---|---|
| [139] | Ca$^{2+}$ | Cation | **Cytotoxicity assay:** Cells were incubated with PLY (0.5–0.025 µg/ml) and Ca 2mM or Ca-free. Minimal toxicity was observed in presence of 2mM Ca whereas in Ca-free buffer, high lytic capacity of PLY was observed even at sublytic concentrations. | **Unknown.** Reduction Ca concentrations improve the PLY membrane binding. Although the mechanism is not fully understood, it could be related to membrane fluidity at different Ca concentrations rather than to a direct binding of the cation to PLY. | | Primary astrocytes from C57BL/6 mice | |
| [140] | Ca$^{2+}$ | Cation | **Cytotoxicity assay:** A549 cells treated with 0.1 µg/mL PLY induces cell death in approximately 50% of the cells after 18 hours as revealed by 2 cellular phenotypes showing either cell death or survival. The surviving population cleared the [Ca2+]m significantly faster than the dying population. Treatment of PLY-induced cells with DMSO reversed the effect, while treatment with the benzothiazepine compound CGP-37 157 (10 µM) (a known Na/Ca exchanger inhibitor) resulted in a decreased number of surviving cells. | **Indirect.** Some alveolar epithelial cells survive apoptosis induction after PLY challenge. Mitochondrial calcium flux is decisive for surviving PLY stimulation. | | A549 cells | |
| [141] | Antibodies | Protein | **Animal models:** Two animal models were used, toxin lethality and pneumonia. Toxin lethality model determined the LD$_{50}$ = 1.06 µg of PLY. 1 h before intranasal infection with *S. pneumoniae*, 100 µg of anti-PLY monoclonal antibodies were intravenously injected. Media survival times was recorded 20 days after infection. The survival rate of mice that received a mix of three monoclonals was 10/20 versus a survival rate of 2/20 in the control group. | **Direct.** Anti-PLY antibodies increased median survival times and survival rates, decreased bacteria number, and leukocytes infiltration. | Pneumonia | MF-1 mice | D39 |
| [68] | Antibodies | Protein | **Animal models:** In slit lamp examinations (SLEs) seven ocular parameters were graded. At 36 and 48 hours after infection by intrastromal injection with *S. pneumoniae*, SLEs were significantly lower in rabbits passively immunized (ear vein injection) with PLY antiserum (IgG titer of 409,600). | **Direct.** Anti-PLY serum elicited protection against severe corneal opacity and massive infiltration of PMNs. | Keratitis | New Zealand White rabbits | WU2 |
| [70] | Antibodies | Protein | **Animal models:** After intranasal inoculation of bacteria to establish nasopharynx colonization, 10 µg or 25 µg of anti-PLY antibodies were administered intravenously. 3 days after inoculation, there was a significant decrease in CFU colonization in mice passively receiving anti-PLY IgG (25 µg). | **Direct.** Anti-PLY antibodies reduce adherence of bacteria to epithelium in vivo (but not in vitro) possibly due to the role of PLY in biofilm formation. | Nasopharyngeal colonization | A549 cells C57BL/6 mice | TIGR4 RX1 |

[1]A549 cells: Human lung epithelial cells

[2]Hep2: Human larynx carcinoma epithelial cells (ATCC CCL-23)

[3]RAW264.7 cells: Macrophage-like, Abelson leukemia virus-transformed cell line derived from BALB/c mice

[4]HBE1

[5]NHBE

[11]H441 cells: Human Bronchial Epithelial cells

[6]HBMEC: Human Brain Microvascular Endothelial Cells

[7]HUVEC: Human Umbilical Vein Endothelial Cells

[8]PMN: Polymorphonuclear Neutrophils

[9]HLMVEC: Human Lung Microvascular Endothelial Cells

[10]HPAEC: Human lung pulmonary artery endothelial cells

[12]HPMEC-ST1.6R: Pulmonary endothelial cell line.

Contrastingly, compared to the total number of trials, there is a high number of antibiotics studies in the context of pneumonia.

## Types of molecules and drugs tested

**Plant-derived compounds: Polyphenols, flavonoids, tannins, quinones, phenethylamines and terpenoids.** The seventeen plant-derived compounds included in this review

belonged to six categories according to their chemical characteristics. Some of them present antibacterial activity [71, 72] and therapeutic effects on acute lung injury since regulating the TLR4/NF-κB signaling pathway, NLRP3 inflammasome activation and the MAPK signaling pathway [73, 74].

Polyphenols are a large and diverse family of naturally occurring organic compounds abundant in plants. They are generally subdivided into phenylpropanoids, flavonoids, hydrolysable tannins, and condensed tannins. In this work, flavonoids and tannins are described independently for greater clarity. An example of polyphenol is Verbascoside, a phenylpropanoid that binds to the cleft between domains 3 and 4 of PLY (Asp471, Asn470, Glu277, Tyr358, and Arg359), as was shown by the molecular dynamic's simulation, thus blocking the conformational transition from monomeric to oligomeric form and inhibiting its lytic activity. Verbascoside was tested in animals suffering from pneumonia with protective effects and reduction of levels of TNF-α and IL-1β [75]. Flavonoids possess the basic structure of a chromone (1,4-benzopyrone) moiety connected to a phenyl ring at position 2. Some examples such as Apigenin [76], Acacetin [77], Amentoflavone [78], and Morin [79] decreased the hemolytic activity of PLY by inhibiting oligomerization. In animal models, the inflammatory cytokines INF-γ and IL-1β in the lungs of mice treated with these compounds decreased significantly. Other flavonoids such as Epigallocatechin gallate (EGCG) significantly increased the survival of infected animals due to it affected PLY oligomerization by binding to the toxin residues Ser256, Glu277, Tyr358, and Arg359 [80]. In addition, this molecule interferes with the biofilm formation and bacterial adherence to cells. Another interesting flavonoid is quercetin, which has been tested in animal models of lung infection, with survival rates of 80% compared to 60% in untreated mice [81]. For other flavonoids such as Dryocrassin ABBA, data from animal models are not available, but *in vitro* results showed relative bactericidal activity and the ability to interfere with toxin oligomerization capacity at concentrations ranging from 0.5 μg/ml to 2 μg/ml [82]. Recently, 27 hydrolysable tannins were tested by Maatsola et al. (2020) [83]. Nanomolar concentrations of Pentagalloylglucose (PGG) and Gemin A were capable of inhibiting the PLY cytolytic capacity. Molecular modeling suggests that PGG also binds to the cleft between domains 2, 3 and 4 (Glu42, Ser256, Asp257, Glu277, and Arg359).

Quinones are a class of organic compounds formally derived from aromatic compounds. Shikonin [84] and Juglone [85] are quinones (naphthoquinones) that inhibited pore formation by means of interfered toxin oligomerization. Aloe-Emodin is a quinone (anthraquinone) with antifungal activity. The authors have proposed this compound together with photodynamic therapy to treat superficial infections of antibiotic-resistant gram-positive bacteria (*Enterococcus faecalis*, *Staphylococcus aureus*, and *S. pneumoniae*). Irradiation generates reactive oxygen species (ROS), leading to the destruction of biomolecules and the killing of bacterial cells; however, the mechanism by which cytotoxins expression (such as PLY) decreases is unknown [71]. Application *in vivo* requires future investigation.

Ephedrine hydrochloride and Pseudoephedrine hydrochloride are phenethylamines from *Ephedra sinica* granules [86] and MXSGT (Ma-xing-shi-gan-tang) [87] are compounds obtained from plants used in traditional Chinese medicine. All of them seem to inhibit the oligomerization of the toxin, although more studies are necessary to specify which residues of PLY are involved. In addition, Ephedrine hydrochloride is part of the WHO model list of essential medicines and is used to prevent low blood pressure during anesthesia, and for the treatment or prevention of attacks of bronchospasm in asthma. It is a very active agonist adrenergic on the receptors of the sympathetic nervous system. Ephedrine hydrochloride increased anti-inflammatory cytokine production IL-10 and decreased the production of pro-inflammatory cytokines TNF-α and IL-12 in dendritic cells in *Staphylococcus aureus*-induced

peritonitis animal models [88]. Current clinical trials are focused on the effect of this compound on sinusitis, rhinitis, etc.

Oleanolic Acid, Betulin and Hederagenin are terpenoids. Oleanolic Acid and its analogues significantly inhibited the activity of important β-lactamases like NDM-1, KPC-2, VIM-1, and OXA-1 as well as the hemolytic action of cytolysins like PLY [89]. For *S. aureus* α-hemolysin, the authors showed through molecular modeling how oleanolic acid can interfere with toxin oligomerization and protect *S. aureus*-infected mice in a combination therapy with β-lactams. More research is needed to see these effects *in vivo* on other toxins, including PLY. Betulin [90] and Hederagenin [91] are triterpenes extracted from plants that interfere with the PLY oligomerization process, without toxic effects against epithelial cells, but have not yet been tested in animal models of pneumococcal infection.

Although most of these above mentioned molecules inhibit or hinder the PLY oligomerization, only three articles [75, 78, 85] studied potential interactions between molecules and toxin by using molecular modelling and docking calculation. Thus, it was probed that the flavonoid Amentoflavone was able to interact with domain 2 (Arg359) and domain 3 (Ser254, Glu277) of the toxin; similarly, to the polyphenol Epigallocatechin gallate, which forms strong interactions with the PLY domain 3 (Ser256, Glu277) and domain 2 (Tyr358, Arg359). Furthermore, Asp 471 from domain 4 of PLY could form a strong hydrogen bond with the hydroxyl group of the benzene ring on the right side of the polyphenol Verbascoside. On the other hand, the interactions of the hydrolysable tannin PGG and PLY take place through flexible galloyl groups of PGG and the pocket formed by domains 2, 3 and 4 of PLY [83]. To our knowledge, no controlled clinical trials have addressed the issue of adjuvant plant-derived compound therapy in the clinical setting of pneumococcal infections. As exception, recent clinical trials using Quercetin (or products containing it) for treating pneumonia in patients with COVID-19 have showed a significantly reduced length of hospitalization, noninvasive oxygen therapy, and number of deaths [92], which clearly confirms its potential as an anti-inflammatory molecule.

**Sterols.** Two sterols, cholesterol and β-sitosterol, inhibited the binding of the toxin to the membrane. Cholesterol is the natural ligand for PLY in the eukaryotic cell membrane. Marquart et al. (2007) [67] showed that cholesterol has a bactericidal effect both *in vitro* and in corneas of rabbits with experimental pneumococcal keratitis. The authors suggest that cholesterol would not only bind to PLY, but also to bacteria and kill them. β-sitosterol is a phytosterol with chemical structure very similar to cholesterol, that has been assayed in murine pneumonia model with good protection results [93]. Additionally, β-sitosterol possess anti-inflammatory properties, causes a dose-dependent inhibition of IL-6 and TNF-α in endotoxin-activated human monocytes, and has beneficial effects on the immune system by increasing the number of viable peripheral blood mononuclear cells and activating the dendritic cells [94, 95]. A detailed study of the binding between five sterols and PLY has confirmed that PLY interacts with these molecules through residues Tyr371, Val372, Leu460, and Tyr461. In the five natural sterols studied, the critical structure is the C22-C23-C24-C25 carbon bonds [58]. We did not find clinical trials of pneumonia related to these sterols.

**Statins.** The protective effect of simvastatin and pravastatin against the cytotoxic effect of PLY *in vitro* was described by Statt et al. (2015) [96, 97]. Oral treatment with simvastatin in pneumonia mice reduced neutrophil infiltration, maintained vascular integrity, and decreased chemokine production [98]. Statins may trigger mevalonate-independent pathways partially via a calcium increase and p38 MAPK activation. Statins act as competitive inhibitors of HMG-CoA reductase (HMGCR), the rate-limiting enzyme of cholesterol synthesis. Simvastatin was also used in a mouse model of sickle cell disease, which is characterized by hemolytic anemia and chronic inflammation, and was shown to lead to a high incidence of invasive pneumococcal pneumonia. The last was due to upregulation of platelet-activating factor

receptor (PAFr). In this context, Simvastatin is useful as it reduced PAFr expression and interfered with toxin pore formation and host cell bacteria invasion [99]. Statins are widely used in the treatment of cardiovascular disease in humans. In last years, statins have begun to be studied in the context of infectious diseases like tuberculosis, AIDS, and COVID-19 [100–102]. Several clinical trials have been carried out with simvastatin in patients with CAP with contradictory results. Some authors find that prior use of this statin improves mortality in patients admitted with CAP [103] while other authors find no difference [104–106]. On the other hand, the administration of 20 mg in individuals admitted with CAP does not show differences in cytokine levels [107]. Recently, clinical trial have been carried out in older adults with derivate CAP bacteremia with encouraging results [108]. Simvastatin could be a good candidate to complement antibiotic therapy. In addition to the bactericidal effect against Gram positive microorganisms, statins have pleiotropic effects on cells of the immune system (i.e., stimulate autophagy in macrophages, increase the number of NK and NKT cells, inhibit MHC-II expression, increase serum levels of IL-10) [108].

**Omega-3 fatty acids.** Omega-3 fatty acids are polyunsaturated fatty acids (PUFAs) characterized by the presence of a double bond, three atoms away from the terminal methyl group in their chemical structure. The structure of docosahexaenoic acid (DHA) is a carboxylic acid with a 22-carbon chain and six *cis* double bonds, with the first double bond located at the third carbon of the omega end. DHA decreased IL-8 PLY mediated in human neutrophils by interfering with $Ca^{2+}$ influx [109]. Also, DHA hinder the binding of toxin to cellular membrane of neutrophils by an unknown mechanism. However, it is accepted that lipid rafts are putative binding sites for cytolysins and that omega-3 polyunsaturated fatty acids exclude proteins from these lipid rafts in eukaryotic cell membranes.

Omega-3 fatty acids are widely known in the field of nutrition and their role is being investigated in neurodegenerative diseases, sleep, etc. As molecules regulating inflammation in the context of infectious diseases, they have been reviewed by Sandhaus & Swick (2021) [110]. The specialized pro-resolving mediators (SPMs) are enzymatically derived from essential fatty acids, including arachidonic acid, eicosapentaenoic acid, and DHA, and have important roles in the resolution of tissue inflammation. DHA is converted to D-series resolvins, protectins, and maresins via 12- and 15-LOX enzymes. DHA-derived SPMs have been shown to be essential for modulating the number and response of T-helper cells to resolve chronic inflammation [111]. Exogenous administration of SPMs in infectious conditions has been shown to be effective at improving infection clearance and survival in preclinical models [112]. Clinical trials carried out with omega-3 fatty acids in the treatment of acute lung injury and acute exacerbations of chronic respiratory disease, asthma, and pneumonia [113, 114], did not show the expected improvement results. However, in patients with COVID-19, C-reactive protein levels, erythrocyte sedimentation rate, and some clinical symptoms were reduced [115].

**Purin-6-ones.** PLY has a known role in increasing the permeability of fluids from microcapillaries to alveolar spaces. Witzenrath et al. (2009) [116] demonstrated the involvement of phosphodiesterase 2 (PDE2) in the hyperpermeability of lung tissue during pneumococcal pneumonia. Purin-6-ones, 9-(6-phenyl-2-oxohex-3-yl)-2-(3,4-dimethoxybenzyl)-purin-6-one (PDP), and hydroxy-PDP are specific inhibitors of PDE2. PDP decreased PLY-induced permeability *in vitro* in endothelial cell cultures and *in vivo* in murine pneumococcal pneumonia models, improving lung damage without interfering with the recruitment of immune system cells into lung tissue. PDE2 enzymes have been also found in various tissues and cells, including pulmonary arterial smooth muscle cells, endothelial cells, platelets, and macrophages. Up to now, there have been no studies available on the use of PDE2 inhibitors in the clinical studies carried out on patients with pneumococcal pneumonia. However, nonselective inhibitors of PDEs as Theophylline has been used in the treatment of bronchial asthma and chronic

obstructive pulmonary disease (COPD) for more than 50 years. Various (selective) PDE3, PDE4, and PDE5 inhibitors have also demonstrated stabilization of the pulmonary epithelial-endothelial barrier and reduction the sepsis- and inflammation-increased microvascular permeability, and suppression of the production of inflammatory mediators, which finally resulted in improved oxygenation and ventilatory parameters [117].

**Thioethers.** Montelukast is a synthetic thioether that acts by decreasing the hyperpermeability and bronchoconstriction produced by the toxin in the lungs [118]. PLY triggers the production of cysteinyl leukotrienes and their receptor CysLT1 in bronchial smooth muscle cells; the interaction of vascular endothelial cells increased vascular permeability, edema, influx of eosinophils, and neutrophils. Montelukast is a CysLT1 antagonist that increases survival in mice with induced pneumococcal pneumonia. In addition, this compound is frequently used as a treatment in patients with asthma and allergies. Recently, montelukast treatment for pneumonia, caused by *Mycoplasma pneumoniae*, and Sars-CoV-2 decreased inflammation markers and contributed to improvements in respiratory insufficiency [119, 120].

**Antibiotics.** Clarithromycin is a derivative of erythromycin and differs structurally in the substitution of an O-methyl group for the hydroxy group at position 6 of the lactone ring. Clarithromycin negatively regulates *ply* gene transcription and activates pneumococcal autolysis [121]. On the other hand, in animal models of pneumonia caused by multidrug-resistant *S. pneumoniae*, the therapy combining levofloxacin and ceftriaxone modulated the inflammatory response (decreased number of neutrophils and polymorphonuclear neutrophils and vascular leakage; decreased nitric oxide production) of pneumococcal infection at sub-MIC levels by downregulating virulence genes (*ply* and *lyt*A), cyclooxygenase-2 (COX-2), and inducible nitric oxide synthase (iNOS) [122]. Generally, macrolides and macrolide-like agents have anti-inflammatory activities, including modulation of cytokine production [123]. Other antibiotics with anti-inflammatory properties are quinolones such as levofloxacin, due to its ability to join to the hydrophobic region of the TLR4-MD2 complex [124]. Clarithromycin is used in the treatment of pneumococcal infections in combination with β-lactams and in some clinical trials using clarithromycin as treatment (revised by Anderson & Feldman, (2017) [59]); the results supported its beneficial role in the anti-inflammatory/immunomodulatory activities. More recently, in a clinical trial in patients with COVID-19, it appears that it could be beneficial for early control of fever and early negative conversion of PCR [125]. However, increasing macrolide resistance in some CAP ethological agents (*S. pneumoniae* and *M. pneumoniae*) and its effect on decreasing the natural T-cell immune response [126], is contributing to reconsideration of their use in the future.

**Peptides.** In this review, seven peptides are presented as candidates for adjuvant therapy. Three of the peptides [64, 65, 69] (J34 binds to the SV1 splice variant of the GHRH receptor, TIP binds PKC-α, and vasculotide binds to the Tie2 angiopeptin receptor) act on endothelium hyperpermeability thus minimizing fluid extravasation into alveolar space. Le et al. (2015) [64] showed how synthetic hybrid peptides, with antimicrobial activity improved survival and reduced the sequelae of lethal pneumococcal systemic infection in animal models. Molecular docking analysis showed that one of the peptides (DM3) had a strong affinity for PLY, also for autolysin and pneumococcal surface protein A (pspA). DM3, in combination with penicillin, showed the recovery of all the treated septicemic animals, but it was not effective in pneumonia, therefore more studies are necessary (Le et al., 2015) [64]. Actually, antimicrobial peptides account for a promising alternative to conventional antibiotics since they can be designed to act on various targets and are less likely to generate resistance [127]. The problems encountered for its clinical application are related to its low absorption and toxicity to human cells. In most cases, these drawbacks can be solved with chemical modifications and *in silico* design [128].

The growth hormone-releasing hormone (GHRH) agonist, JI-34, decreased PLY-mediated endothelial hyperpermeability by reducing the phosphorylation of myosin light chain and vascular endothelial (VE)-cadherin and restoring basal levels of $Na^+$ [129]. Another example is Vasculotide (a tetrameric peptide with four octapeptides ($NH_3$-CHHHRHSF-COOH), which are covalently attached through $NH_2$-terminal maleimide) which improved the barrier function of the endothelial epithelium during pneumococcal pneumonia (Gutbier et al., 2017) [130]. The action was mediated by its binding to the Tie2 receptor of angiopeptin (Ang/Tie) thereby protecting against increased levels of fluid invading the intra-alveolar space and further pulmonary edema. Other drugs targeting the Ang/Tie pathway are being tested for the control of pulmonary vascular disorders in clinical trials. AV-001, a Vasculotide-derived peptide (Vasomune Therapeutics) is currently in phase I. The results showed that AV-001 is safe and well tolerated in patients with COVID-19, acute respiratory distress syndrome (ARDS), or COVID-19-associated ARDS [131].

The TIP peptide (TNF-derived tonoplast intrinsic protein) and the Ro 32–0432 peptide (bisindolylmaleimide XI hydrochloride) are Protein Kinase C-α (PKC-α) inhibitors. PKC-α is stimulated by the $Ca^{2+}$ influx mediated through PLY-induced pores which increased RhoA / RKK; arginase I reduces NO generation *in vivo* and consequently, microvessel leakage. TIP peptide diminished PKC-α activation and therefore, reduced PLY-induced endothelial hyperpermeability [129]. AP301 (Solnatide) is a circularized derivative of TIP peptide and has a Phase IIB clinical trial in ARDS [132] and is also being tested in patients with COVID [132]. The expected results, based on the previous studies, are activation of the pulmonary sodium ion channel (ENaC) to directly activate alveolar liquid clearance and reduce the leakage of blood and fluids from the capillaries in the airspace, i.e., accelerate the resolution of alveolar edema and reduce barrier injury in the lung.

Among the therapeutic peptides studied, some of them were designed to form hydrogen bridges with the cholesterol-interaction loop of different CDCs (PLY, LLO, and SLO) in a way that inhibits their hemolysis capacity and their pro-inflammatory activity. Thus, the named MRC-1 peptides bind PLY in domain 4 and inhibit binding to the human mannose receptor (MRC-1/CD206, a phagocytic receptor for bacteria, fungi, and other pathogens) [65]. Therefore, they reduced PLY-induced damage to the integrity of the epithelial barrier, the release of the chemokine IL-8, and the pro-inflammatory cytokines TNF-α and IL-12 as well as blocked bacterial invasion into the epithelium. Another example includes a peptide generated with the 70 C-terminal amino acids of PLY (C70PLY), which competed with MD2 to bind to TLR-4, inhibiting the effect of lipopolysaccharide on the phosphorylation of the ERK1/2 and NFκB-p65 subunit. TLR-4 is a member of the toll-like receptor family that senses pathogen-associated molecular patterns, stimulating cytokine production and apoptosis. Also, its activation leads to an intracellular signaling pathway NF-κB and the activation of the innate immune system [69].

As mentioned above, the innate immune system recognizes the pathogen components through TLR receptors, which trigger the host's first defense system. The inflammatory action of PLY is mediated by TLR-4 and induces caspase 3-dependent apoptosis in macrophages. zVAD (carbobenzoxy-valyl-alanyl-aspartyl-[O-methyl]-fluoromethylketone) is a modified peptide caspase inhibitor used to block apoptosis [31, 133]. However, apoptosis appears to be a defense mechanism against invasion by pneumococcus; therefore, inhibition of apoptosis results in increased mortality in animal models of pneumococcal pneumonia. Yet, cells can recover from PLY-induced apoptotic cell death by controlling mitochondrial $[Ca^{2+}]_m$ flux which contributes to the preservation of membrane potential (ΔΨm). PLY binding cholesterol in rich cellular membranes leads to large pores (250–350 Å) which contributes to a rapid influx of $Ca^{2+}$ into the cytoplasm. $Ca^{2+}$ induces repair mechanisms for protection of the host cell

[134]. One of them is sequestering $Ca^{2+}$ into the mitochondria. However, exceeding the mitochondrial capacity can result into release of pro-apoptotic factors such as cytochrome C (cytC) or apoptosis inducing factor (AIF) [26]. We did not find clinical trials related to with the treatment of diseases with zVAD.

**Cations.** Cations like $Ca^{2+}$, $Mg^{2+}$, and $Zn^{2+}$ hinder pore formation thereby decreasing the cytolytic capacity of PLY. Hupp et al. (2017) [66] showed that $Mg^{2+}$ decreases the pore-forming capacity of PLY in primary glial cells, improving survival in animal models of meningitis. Cations are known to influence membrane fluidity or inhibit toxin binding, but these hypotheses did not hold for $Mg^{+2}$ since it does not prevent the binding of the toxin to the membrane, nor does it seem to change its characteristics. Another divalent cation, $Zn^{2+}$ [135], protected rat cochlear hair cells from the toxic action of PLY by preventing its binding to the membrane. Recently, the deficiency of $Zn^{2+}$ has been involved in a higher risk of acute infection in nasopharyngeal *S. pneumoniae* carriers. Therefore, supplementation with this element would improve immune system performance and consequently reduce pneumonia [136], Recent research on the relationship between micronutrients and some infections has shown improved immune function [137, 138]. Unlike the other cations, Wippel et al. (2011) [139], using brain tissue, postulated that $Ca^{2+}$ could vary the fluidity of the membrane rather than having a direct action on the toxin because reductions in $Ca^{2+}$ concentrations improve the binding of the PLY membrane thus increasing its lytic capacity. However, *S. pneumoniae* is capable of inducing apoptosis thanks to the pores formed by PLY which produce rapid inflows of mitochondrial calcium $[Ca^{2+}]_m$ thereby favoring fragmentation, loss of motility, and membrane potential in addition to activation of caspase 3. Nerlich et al., (2021) [140] showed that a significant number of alveolar epithelial cells survived caspase activation after being challenged with PLY, critically regulating $[Ca^{2+}]_m$ to control cell fate after attack by PLY. Therefore, calcium ions were postulated to be a useful candidate in therapeutic intervention during pneumococcal infection.

**Antibodies.** Antibodies against PLY have been used against pneumonia, keratitis, and nasopharyngeal colonization in animal models of infection [68, 70, 141]. The protective effect of mouse monoclonal antibodies in reducing PLY proinflammatory properties was clearly demonstrated as well as an increased survival rate with decreased bacteria lungs colonization, leukocyte infiltration, and lung injury [141]. Similarly, a reduction in toxin effects was observed in an induced keratitis model in rabbits using anti-PLY polyclonal antiserum [68]. Interestingly, the efficacy of human anti-PLY antibodies in reducing *S. pneumoniae* nasopharyngeal colonization was also demonstrated in mice [70]. Although PLY is an intracellular protein that is released during bacteria lysis, it would also be located on the cell surface, where it would play a role in the aggregation and formation of biofilms. Potential rejection reactions to repeated doses of antibodies have been solved through the humanization of monoclonal antibodies, which has facilitated their use for the treatment of various diseases [142]. This, together with the fact that immunoglobulins have been used since ancient times for the treatment of infectious diseases, makes this strategy very promising. In fact, the effect of PLY on platelets and erythrocytes can be neutralized by polyvalent human IgG and trimodulin [143]. In a clinical trial in patients with *S. pneumoniae* CAP, treatment with trimodulin (182.6 mg/kg, for 5 consecutive days) improved survival compared to patients treated with placebo [45]. Drugs developed against toxins from other microorganisms, including monoclonal antibodies, antibody fragments, antibody mimetics, have recently been reviewed [144]. It is important to highlight the phase I clinical trial using liposomes to capture *S. pneumoniae* toxins [145].

**Network of compounds and related proteins.** In order to evaluate the molecular interactions of the 39 compounds with neutralizing toxin properties described in the selected articles, the EXTRACT 2.0 tool was used for text mining of biomedical named entities and ontology terms such as diseases, subcellular localizations, tissues, drugs, and other small molecule

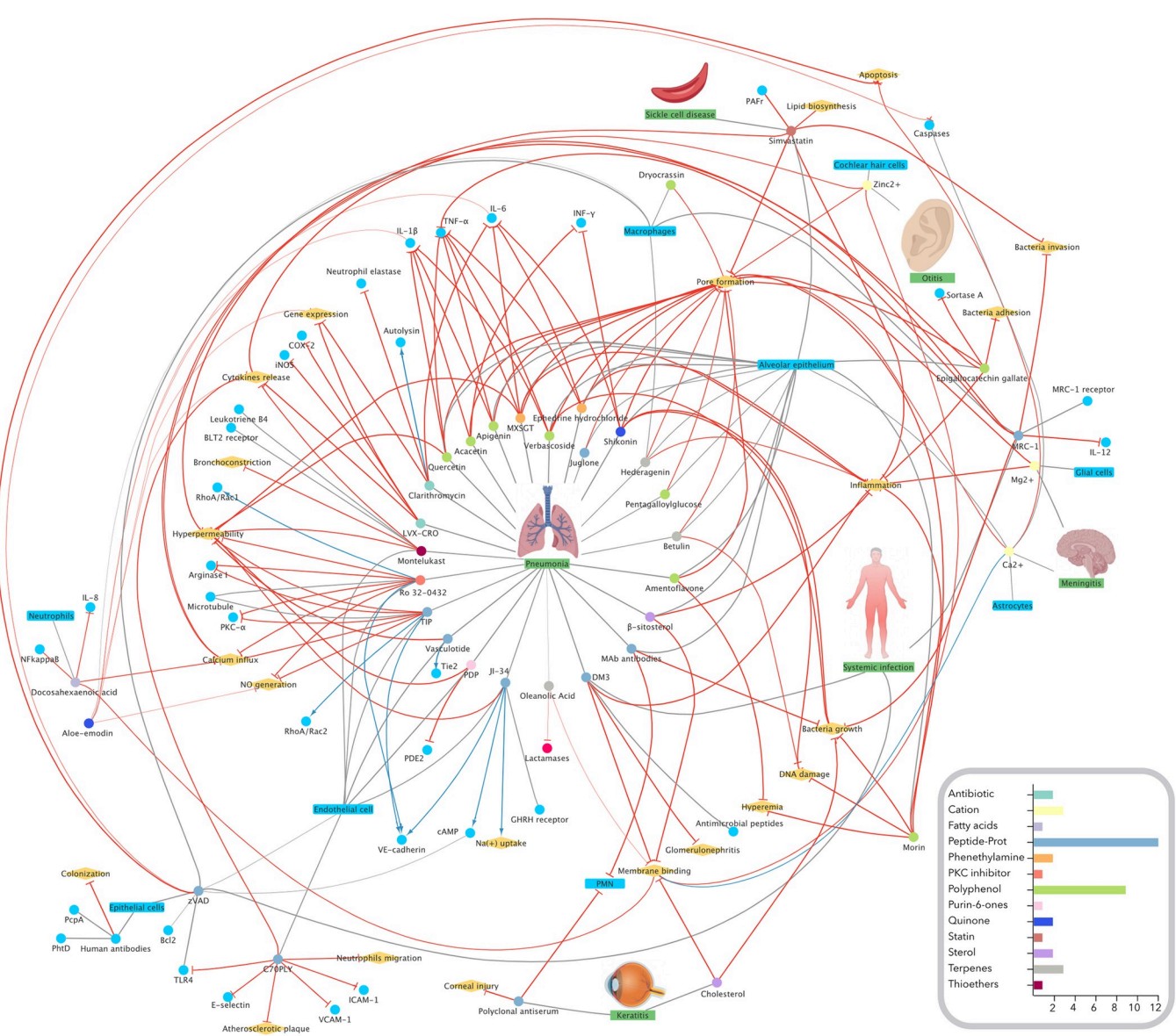

**Fig 3. Knowledge network of the protective molecules against PLY.** PLY interactions (edges) with host molecules (cyan circles), cells (cyan rectangles), and cellular processes (orange diamonds) is shown. Three types of connections are distinguished: the red arrows indicate inhibition (⊥) whereas the blue arrows refer to activation (↑) and the grey arrows show interactions without inhibition or activation. The width of the lines reflects article ranking. The right panel represents the abundance and color of the molecules grouped by type. The network was built using Cytoscape software after text mining articles with EXTRACT 2.0.

compounds [62]. With the resulting dataset, a knowledge network (**Fig 3**) was built using Cytoscape (see Materials and methods). Analysis of the network reveals that the most affected process is the formation of the pore with 19 interactions (edges), followed by hyperpermeability and inflammation (showing nine interactions), and membrane binding (seven interactions). The relevance and complexity of pneumonia is evidenced by the high number of molecules (24) used for its treatment while other diseases show three or fewer interactions with therapeutic molecules. Regarding host cell proteins, TNF-α was the protein most affected with 6 interactions, followed by IL-6 and, IL-1β (both showing five interactions), and VE-cadherin (three interactions). The presented knowledge network visually shows the summary of

the reviewed articles. In addition to this, the structure of the network reveals the molecules with the greatest interactions which can be useful in the search for new treatments for the neutralization of the toxin. Drug repurposing identifies new uses of drugs outside their original medical scope, considering that, sometimes, one molecule can act on multiple targets [146]. Recently, a network medicine platform based on systems pharmacology was used to reposition antiviral drugs against SARS-CoV-2 [147]. This approach aims to reduce both development costs and the time it takes for already approved drugs to reach the market.

## Current limitations and future perspectives

Despite the effective development of antibiotics and more recently pneumococcal vaccines, pneumococcal pneumonia continues to cause high morbidity, even though actually the mortality caused to invasive pneumococcal disease is declining due to childhood vaccination and herd protection [148]. The aging of the population and the combination of pneumococcal disease with other important viruses such as HIV, influenza and, more recently, SARS-CoV-2, could be behind these high levels of mortality and morbidity. To proceed in the discovery of new strategies to combat this disease, we decided to focus our work on PLY and its role in the performance of pneumococcal infections.

The combination of antibiotics with other molecules that can somehow block the action of PLY, was presented as a hopeful therapy aimed at significantly reducing the severity of pneumococcal pneumonia and therefore, diminishing the high mortality that is still associated with it. The search strategy for the review was broad to locate all studies in which some type of molecule was used as an alternative treatment or blockade of the toxin. The articles reviewed focused on experimental and laboratory studies and provide reduced information about the behavior with other doses or routes of inoculation. Most studies do not provide information on survival rates and limit themselves to studying the characteristics of the organs or tissues involved. One of the limitations of this systematic review is the lack of homogeneity of the studies. We did not find comparable studies in molecules, doses, inoculation routes, etc. that allow a statistical comparison of the results or meta-analysis. However, most of the studies reviewed has a good level of confidence. The clinical efficacy of the molecules in the treatment of pneumococcal infections will depend on several factors such as the pharmacokinetic/pharmacodynamic parameters of the molecule used for treating the infection, and the infection site (i.e. Central Nervous System is a difficult place for the molecules to penetrate). Indeed, some compounds have a good perspective for inhibition of pore formation and apoptosis, such as polyphenols and sterols, which also exhibit anti-inflammatory properties. With respect to hyperpermeability, it appears that it can be reduced by thioethers and some antimicrobial peptides. However, although some of these compounds have a great potential to be implemented as adjuvants in the treatment of pneumococcal infections, others (like peptides revised in this study), require future clinical trials using different combinations and deeper investigation. Hopefully, recent research related to the SARS-CoV-2 pandemic will help to understand the effect of these molecules with respect to ARDS in pneumococcal pneumonia.

Other limitation of this review was that surprisingly, for some molecules, (like cations and cholesterol), of which we did not find PLY-related studies in animal models, there are a considerable number of clinical trials studies related with pneumonia. We considered that this need exhaustive testing and are out of score of this review.

In summary, although the clinical relevance of this review is limited due to the lack of clinical tests in humans, some molecules have already been clinically tested and therefore could be safe for use in pneumococcal infections.

## Supporting information

**S1 Fig. Pneumolysin (PLY) action mechanisms in the different body compartments.** (a) Toxin interactions with cellular membranes (I), human immune system (II), and endothelium (III) are depicted. (b) The pneumococcus spreading routes and the main protein domains relevant to the structure of PLY are indicated.
(TIF)

**S1 Table. PRISMA 2020 checklist.**
(DOCX)

**S2 Table. Data extraction items.**
(DOCX)

**S3 Table. Study quality assessment based on the OHAT risk assessment.**
(DOCX)

**S4 Table. Articles and number of clinical trials.**
(DOCX)

**S5 Table. Chemical structures of molecules with effects against PLY.**
(DOCX)

## Acknowledgments

We acknowledge Dr. Pilar García (IPLA, CSIC, Spain) for critical review of manuscript.

## Author Contributions

**Conceptualization:** Felipe Molina, María del Mar García-Suárez.

**Data curation:** José Ignacio López-Sánchez, Efrén Pérez-Santín.

**Formal analysis:** Felipe Molina, José Ignacio López-Sánchez, Efrén Pérez-Santín, María del Mar García-Suárez.

**Funding acquisition:** María Dolores Cima Cabal.

**Investigation:** Felipe Molina.

**Software:** Felipe Molina.

**Supervision:** María del Mar García-Suárez.

**Writing – original draft:** María Dolores Cima Cabal, Felipe Molina, José Ignacio López-Sánchez, María del Mar García-Suárez.

**Writing – review & editing:** María Dolores Cima Cabal, Efrén Pérez-Santín, María del Mar García-Suárez.

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
