## [Decision Letter · Decision Letter 0]

2 Jan 2023

PONE-D-22-26630Pneumolysin as a target for new therapies against pneumococcal infections: A systematic reviewPLOS ONE

Dear Dr. Mar,

Thank you for submitting your manuscript to PLOS ONE. After careful consideration, we feel that it has merit but does not fully meet PLOS ONE’s publication criteria as it currently stands. Therefore, we invite you to submit a revised version of the manuscript that addresses the points raised during the review process.

We look forward to receiving your revised manuscript.

Kind regards,

Raj Kumar Koiri

Academic Editor

PLOS ONE

Journal Requirements: 

" ext-link-type="uri" xlink:type="simple">https://journals.plos.org/plosone/s/file?id=ba62/PLOSOne_formatting_sample_title_authors_affiliations.pdf"

"No, The funders had no role in study design, data collection and analysis, decision to publish, or preparation of the manuscript."

"NO authors have competing interests"

 This information should be included in your cover letter; we will change the online submission form on your behalf

Reviewers' comments:

Reviewer's Responses to Questions

**Comments to the Author**

1. Is the manuscript technically sound, and do the data support the conclusions?

Reviewer #1: Yes

Reviewer #2: Partly

Reviewer #3: No

2. Has the statistical analysis been performed appropriately and rigorously? 

Reviewer #1: N/A

Reviewer #2: Yes

Reviewer #3: N/A

3. Have the authors made all data underlying the findings in their manuscript fully available?

Reviewer #1: Yes

Reviewer #2: Yes

Reviewer #3: Yes

4. Is the manuscript presented in an intelligible fashion and written in standard English?

Reviewer #1: No

Reviewer #2: Yes

Reviewer #3: No

5. Review Comments to the Author

Reviewer #1: I read your article with great interest and would like to present forward the following comments/ suggestions:

1) Study selection paragraph -

a) Possible typing error in line 239 -- will be 2 case reports, not 3 (as per PRISMA diagram)

b) Lines 241-244 -- Out of 141 articles 108 were excluded and 8 articles from citation were added to select total 41 articles, however complete breakup of these 108 articles are not given. (break-up of 100 articles excluded are given, 66+31+3)

2) It is clear that there was considerable heterogeneity between all the studies selected. Also, there were no comparison studies. Both of these indicate that the level of evidence generated by these studies would be on the lower side. Hence, there was no statistical analysis done and neither were there any conclusions drawn regarding the topic related to clinical decision making.

Nevertheless, it was an interesting read and underlines that a lot of ground needs to be covered in this area before we can apply it to clinical practice.

Reviewer #2: The authors performed a systematic review on pneumolysin as a target for therapies in pneumococcal infections.

The authors should consider adding conclusion section other than discussion section in the abstract.

The authors should discuss in the introduction section the clinical relevance of their study specially how it is compared to the current standard of clinical practice in managing pneumococcal infection (antibiotics and vaccines).

Despite the authors stating in the discussion section that mortality is high from pneumococcal infection, actually the mortality is declining due to childhood vaccination, please see reference below:

Grau I, Ardanuy C, Cubero M, Benitez MA, Liñares J, Pallares R. Declining mortality from adult pneumococcal infections linked to children's vaccination. J Infect. 2016;72(4):439-449. doi:10.1016/j.jinf.2016.01.011

Would consider adding more points in the discussion about the presumable effects of these compounds and the relevance to pneumococcal infection as it is more focused on their effect in COVID.

I commend the authors for the effort they put in this study, but I would like to see more points about the clinical relevance of this review given that the included studies are mainly animal experiments.

Reviewer #3: The Review article targets one of the most important virulence factor of pneumococci and summarizes potential intervention strategies to neutralize the mode of action of pneumolysin. The Review is thereby focusing on plant-derived compounds, although antibodies and there inhibitory effect are also discussed and listed in table 1.

The Review is in time and important, because pneumolysin and it´s cytolytic and cytotoxic effects are a major driver in the pathogenesis of this pathogen. Blocking pneumolysin and preventing binding to the host cell or oligomerization prior to pore-formation are therefore perfect mechanism to combat severe pneumococcal invasive infections.

Overall, the reviewer is not convinced about the structure, because reading is not enjoyable despite the topic is of high interest. The separation of Results and Discussion is not beneficial in the view of the reviewer. In particular when the Review starts with the “Types of molecules and drugs tested”, the discussion can be included and combined with the summarized studies. This would result in a nice flow and avoid redundancies.

A major flaw of the Review is the partially inaccurate literature research, which is quite obvious in the introduction (see a few examples in the point by point comments) and a critical issue.

In addition, the linguistic expression and English can be improved (examples: lines 115, 147 -149, 152 (the brain does not react – this is an inflammation of the meninges), 373 (possess and causes), 515 (protector? (protective) effect), 576 (when starting with Regarding then the sentence is incomplete (similar to other sentences)), line 600 (Mycoplasma pneumoniae), line 633 and others

Major comments:

1. Pneumolysin is a CDC but also binds according to the literature to MRC-1 (see e.g. doi: 10.1038/s41564-018-0280-x, doi:10.15252/emmm.202012695.) and this should be mentioned in the Introduction

2. Serotype 1 and 8 with mild infections is not overall correct; see newer publications:

doi: 10.1038/s41467-020-15751-6.

doi: 10.1016/j.tim.2021.11.007.

3. It is not correct that oligomerization is with up to 400 and 500 toxin monomers (the pore is a 400 Å); these are approx. 42 monomers – see publications (e.g. doi: 10.7554/eLife.23644)

4. Line 121: influx of cytosolic calcium?

5. Line 125: sensing TLR4 is under debate, this has to be mentioned because of contradictory results

6. Line 159: the authors are not familiar with the newest publications; the statement is not correct anymore (see doi: 10.1182/bloodadvances.2020002372)

7. Combining results and discussion will be highly beneficial for this Review

8. When the effect of antibodies are discussed, the authors have also to consider these studies: doi: 10.1182/bloodadvances.2020002372, doi: 10.1055/a-1723-1880

Minor comments:

1. Line 94: change LPXTG proteins to “sortase-anchored proteins

2. Line 99: Re-check your statement in line 99-100 that pneumolysin is expressed in the late-log phase. This is not correct, instead PLY is produced constitutively and the free or released PLY is higher in the late log phase because of autolysis and the heterogeneity in the culture

3. Line 103: again, this is not exact what happens; re-write after checking the literature

6. PLOS authors have the option to publish the peer review history of their article (what does this mean?). If published, this will include your full peer review and any attached files.

Reviewer #1: **Yes: **Dr. Saikat Banerjee, MD Respiratory Medicine, ERS HERMES Diplomate, Micromasters in Statistics and Data Science from MIT

Reviewer #2: No

Reviewer #3: **Yes: **Sven Hammerschmidt

---

## [Author Response · Author response to Decision Letter 0]

23 Jan 2023

Reviewer #1: I read your article with great interest and would like to present forward the following comments/ suggestions:

1) Study selection paragraph –

a) Possible typing error in line 239 -- will be 2 case reports, not 3 (as per PRISMA diagram)

b) Lines 241-244 -- Out of 141 articles 108 were excluded and 8 articles from citation were added to select total 41 articles, however complete breakup of these 108 articles are not given. (break-up of 100 articles excluded are given, 66+31+3)

We agree with the reviewer and have fixed both errors: lines 275 and 278. We thank the reviewer for the meticulous review of the manuscript.

2) It is clear that there was considerable heterogeneity between all the studies selected. Also, there were no comparison studies. Both of these indicate that the level of evidence generated by these studies would be on the lower side. Hence, there was no statistical analysis done and neither were there any conclusions drawn regarding the topic related to clinical decision making.

Nevertheless, it was an interesting read and underlines that a lot of ground needs to be covered in this area before we can apply it to clinical practice.

We appreciate the work of the reviewer and the comments to improve the manuscript.

 

Reviewer #2: The authors performed a systematic review on pneumolysin as a target for therapies in pneumococcal infections.

The authors should consider adding conclusion section other than discussion section in the abstract.

In accordance with the reviewer's comment, we have added a Conclusion section to the Abstract.

The authors should discuss in the introduction section the clinical relevance of their study specially how it is compared to the current standard of clinical practice in managing pneumococcal infection (antibiotics and vaccines).

The authors agree with the reviewer's comment and have added the following text in Introduction:

“Current pneumococcal conjugate vaccines are a 13-valent pneumococcal conjugate vaccine (PCV13) for routine pediatric immunization and a 23-valent pneumococcal polysaccharide vaccine (PPSV23) for adults aged ≥65 years. Since 2014, PCV13 was also recommended for all adults aged ≥65 years (Matanock et al., 2020). In 2021, two new vaccines have been approved by the FDA for use in adults ≥18 years (PCV-15 and PCV-20) that are currently under evaluation (Kobayashi et al., 2022). These vaccines are immunogenic and effective and prevent disease caused by the serotypes whose capsule types are in the vaccine. However, these vaccines do not cover the full spectrum of invasive pneumococcal serotypes. On the other hand, the management of pneumococcal infections in clinical practice involves the use of broad-spectrum antibiotics, usually a combined therapy of b-lactams and macrolides (Metlay et al., 2019). However, S. pneumoniae has developed resistance to multiple antibiotics including penicillin, macrolides, fluoroquinolone, and sulfamethoxazole-trimethoprim (Li et al., 2023). The emergence of non-vaccine serotypes after the introduction of PCV, together with increased antibiotic resistance in these serotypes, has become a global threat. (Lo et al., 2019). Taking together the development of new therapies is necessary for the effective treatment of pneumococcal infections”. (lines 182-196)

Matanock A, Lee G, Gierke R, Kobayashi M, Leidner A, Pilishvili T. Use of 13-Valent Pneumococcal Conjugate Vaccine and 23-Valent Pneumococcal Polysaccharide Vaccine Among Adults Aged ≥65 Years: Updated Recommendations of the Advisory Committee on Immunization Practices. MMWR Morb Mortal Wkly Rep. 2019 Nov 22;68(46):1069-1075. doi: 10.15585/mmwr.mm6846a5. Erratum in: MMWR Morb Mortal Wkly Rep. 2020 Jan 03;68(5152):1195. 

Kobayashi M, Farrar JL, Gierke R, Britton A, Childs L, Leidner AJ, Campos-Outcalt D, Morgan RL, Long SS, Talbot HK, Poehling KA, Pilishvili T. Use of 15-Valent Pneumococcal Conjugate Vaccine and 20-Valent Pneumococcal Conjugate Vaccine Among U.S. Adults: Updated Recommendations of the Advisory Committee on Immunization Practices - United States, 2022. MMWR Morb Mortal Wkly Rep. 2022 Jan 28;71(4):109-117. doi: 10.15585/mmwr.mm7104a1. 

Metlay JP, Waterer GW, Long AC, Anzueto A, Brozek J, Crothers K, Cooley LA, Dean NC, Fine MJ, Flanders SA, Griffin MR, Metersky ML, Musher DM, Restrepo MI, Whitney CG. Diagnosis and Treatment of Adults with Community-acquired Pneumonia. An Official Clinical Practice Guideline of the American Thoracic Society and Infectious Diseases Society of America. Am J Respir Crit Care Med. 2019 Oct 1;200(7):e45-e67. doi: 10.1164/rccm.201908-1581ST. 

Li L, Ma J, Yu Z, Li M, Zhang W, Sun H. Epidemiological characteristics and antibiotic resistance mechanisms of Streptococcus pneumoniae: An updated review. Microbiol Res. 2023 Jan;266:127221. doi: 10.1016/j.micres.2022.127221. 

Lo SW, Gladstone RA, van Tonder AJ, Lees JA, du Plessis M, Benisty R, Givon-Lavi N, Hawkins PA, Cornick JE, Kwambana-Adams B, Law PY, Ho PL, Antonio M, Everett DB, Dagan R, von Gottberg A, Klugman KP, McGee L, Breiman RF, Bentley SD; Global Pneumococcal Sequencing Consortium. Pneumococcal lineages associated with serotype replacement and antibiotic resistance in childhood invasive pneumococcal disease in the post-PCV13 era: an international whole-genome sequencing study. Lancet Infect Dis. 2019 Jul;19(7):759-769. doi: 10.1016/S1473-3099(19)30297-X. 

Despite the authors stating in the discussion section that mortality is high from pneumococcal infection, actually the mortality is declining due to childhood vaccination, please see reference below:

Grau I, Ardanuy C, Cubero M, Benitez MA, Liñares J, Pallares R. Declining mortality from adult pneumococcal infections linked to children's vaccination. J Infect. 2016;72(4):439-449. doi:10.1016/j.jinf.2016.01.011

In accordance with the reviewer's suggestion, we have modified the text and inserted the reference: “Despite the effective development of antibiotics and more recently pneumococcal vaccines, pneumococcal pneumonia continues to cause high morbidity, even though actually the mortality caused to invasive pneumococcal disease is declining due to childhood vaccination and herd protection (Grau et al., 2016)” (lines 677-680)

Would consider adding more points in the discussion about the presumable effects of these compounds and the relevance to pneumococcal infection as it is more focused on their effect in COVID.

Although for most of the molecules we did not find clinical trials in pneumococcal infections, regarding the presumed effects of these compounds, most of them seem to have anti-inflammatory properties, others protective effects on the endothelial barrier, expected to decrease lung damage and edema. We believe that more studies should be carried out in order to test their efficacy; however, the fact that some of them are already used in the clinic suggests that they might be safe in humans.

In accordance with the reviewer's suggestions, we have added the following text in the Current limitations and future perspectives section: 

“The clinical efficacy of the molecules in the treatment of pneumococcal infections will depend on several factors such as the pharmacokinetic/pharmacodynamic parameters of the molecule used for treating the infection, and the infection site (i.e. Central Nervous System is a difficult place for the molecules to penetrate)” (lines 698-701)

In accordance with the reviewer's suggestions, we have added the following text in the Results and Discussion section: 

“PDE2 enzymes have been also found in various tissues and cells, including pulmonary arterial smooth muscle cells, endothelial cells, platelets, and macrophages. Up to now, there have been no studies available on the use of PDE2 inhibitors in the clinical studies carried out on patients with pneumococcal pneumonia. However, nonselective inhibitors of PDEs as Theophylline has been used in the treatment of bronchial asthma and chronic obstructive pulmonary disease (COPD) for more than 50 years. Various (selective) PDE3, PDE4, and PDE5 inhibitors have also demonstrated stabilization of the pulmonary epithelial-endothelial barrier and reduction the sepsis- and inflammation-increased microvascular permeability, and suppression of the production of inflammatory mediators, which finally resulted in improved oxygenation and ventilatory parameters (Mokra and Mokry, 2021)” (lines 480-490)

Mokra D, Mokry J. Phosphodiesterase Inhibitors in Acute Lung Injury: What Are the Perspectives? Int J Mol Sci. 2021 Feb 16;22(4):1929. doi: 10.3390/ijms22041929. 

“Several clinical trials have been carried out with Simvastatin in patients with CAP with contradictory results. Some authors find that prior use of this statin improves mortality in patients admitted with CAP (Chalmers et al., 2008; Mortesen et al., 2008; Thomsen et al., 2008) while other authors find no difference (Majumdar et al., 2006). On the other hand, the administration of 20 mg in individuals admitted with CAP does not show differences in cytokine levels (Viasus et al., 2015)” (lines 435-440)

Chalmers JD, Singanayagam A, Murray MP, Hill AT. Prior statin use is associated with improved outcomes in community-acquired pneumonia. Am J Med. 2008 Nov;121(11):1002-1007.e1. doi: 10.1016/j.amjmed.2008.06.030. 

Mortensen EM, Pugh MJ, Copeland LA, Restrepo MI, Cornell JE, Anzueto A, Pugh JA. Impact of statins and angiotensin-converting enzyme inhibitors on mortality of subjects hospitalized with pneumonia. Eur Respir J. 2008 Mar;31(3):611-7. doi: 10.1183/09031936.00162006. 

Thomsen RW, Riis A, Kornum JB, Christensen S, Johnsen SP, Sørensen HT. Preadmission use of statins and outcomes after hospitalization with pneumonia: population-based cohort study of 29,900 patients. Arch Intern Med. 2008 Oct 27;168(19):2081-7. doi: 10.1001/archinte.168.19.2081. 

Majumdar SR, McAlister FA, Eurich DT, Padwal RS, Marrie TJ. Statins and outcomes in patients admitted to hospital with community acquired pneumonia: population based prospective cohort study. BMJ. 2006 Nov 11;333(7576):999. doi: 10.1136/bmj.38992.565972.7C. 

Viasus D, Garcia-Vidal C, Simonetti AF, Dorca J, Llopis F, Mestre M, Morandeira-Rego F, Carratalà J. The effect of simvastatin on inflammatory cytokines in community-acquired pneumonia: a hospitaliz, double-blind, placebo-controlled trial. BMJ Open. 2015 Jan 6;5(1):e006251. doi: 10.1136/bmjopen-2014-006251. 

I commend the authors for the effort they put in this study, but I would like to see more points about the clinical relevance of this review given that the included studies are mainly animal experiments.

We thank the reviewer for the comments and suggestions. We concur, the clinical relevance of this work is limited due to the lack of studies in humans. To acknowledge this problem, we have added the following text: 'In summary, although the clinical relevance of this review is limited due to the lack of clinical tests in humans, some molecules have already been clinically tested and therefore could be safe for use in pneumococcal infections' (lines 711-713)

Nevertheless, some of the molecules have been tested in patients with ADRSL (Solnatide, AV-001), CAP (clarithromycin; Simvastatin, omega-3) or Sars-CoV-2 pneumonia (Montelukast); others are used for other purposes in humans (ß-sitosterol, omega-3), so they could be safe for use in pneumococcal pneumonia in a future.

 

Reviewer #3: The Review article targets one of the most important virulence factor of pneumococci and summarizes potential intervention strategies to neutralize the mode of action of pneumolysin. The Review is thereby focusing on plant-derived compounds, although antibodies and there inhibitory effect are also discussed and listed in table 1.

The Review is in time and important, because pneumolysin and it´s cytolytic and cytotoxic effects are a major driver in the pathogenesis of this pathogen. Blocking pneumolysin and preventing binding to the host cell or oligomerization prior to pore-formation are therefore perfect mechanism to combat severe pneumococcal invasive infections.

Overall, the reviewer is not convinced about the structure, because reading is not enjoyable despite the topic is of high interest. The separation of Results and Discussion is not beneficial in the view of the reviewer. In particular when the Review starts with the “Types of molecules and drugs tested”, the discussion can be included and combined with the summarized studies. This would result in a nice flow and avoid redundancies. A major flaw of the Review is the partially inaccurate literature research, which is quite obvious in the introduction (see a few examples in the point by point comments) and a critical issue.

In addition, the linguistic expression and English can be improved (examples: lines 115, 147 -149, 152 (the brain does not react – this is an inflammation of the meninges), 373 (possess and causes), 515 (protector? (protective) effect), 576 (when starting with Regarding then the sentence is incomplete (similar to other sentences)), line 600 (Mycoplasma pneumoniae), line 633 and others.

Major comments:

1. Pneumolysin is a CDC but also binds according to the literature to MRC-1 (see e.g. doi: 10.1038/s41564-018-0280-x, doi:10.15252/emmm.202012695.) and this should be mentioned in the Introduction.

In accordance with the suggestion, the text has been modified and suggested references added: “PLY is a toxin that binds eukaryotic membrane cholesterol (belongs to cholesterol-dependent cytolysins, CDC) but also binds to mannose receptor C type 1 (MRC-1) promoting an anti-inflammatory response and reducing pneumococcal disease (Subramanian et al., 2019; 2020). In this way, this toxin has double functionality (“sword and shield” or “Yin and Yang”) (Surve et al., 2018; Pereira et al., 2022)” (lines 99-102).

Subramanian K, Neill DR, Malak HA, Spelmink L, Khandaker S, Dalla Libera Marchiori G, Dearing E, Kirby A, Yang M, Achour A, Nilvebrant J, Nygren PÅ, Plant L, Kadioglu A, Henriques-Normark B. Pneumolysin binds to the mannose receptor C type 1 (MRC-1) leading to anti-inflammatory responses and enhanced pneumococcal survival. Nat Microbiol. 2019 Jan;4(1):62-70. doi: 10.1038/s41564-018-0280-x. 

Subramanian K, Iovino F, Tsikourkitoudi V, Merkl P, Ahmed S, Berry SB, Aschtgen MS, Svensson M, Bergman P, Sotiriou GA, Henriques-Normark B. Mannose receptor-derived peptides neutralize pore-forming toxins and reduce inflammation and development of pneumococcal disease. EMBO Mol Med. 2020 Nov 6;12(11):e12695. doi: 10.15252/emmm.202012695. 

Pereira JM, Xu S, Leong JM, Sousa S. The Yin and Yang of Pneumolysin During Pneumococcal Infection. Front Immunol. 2022 Apr 22;13:878244. doi: 10.3389/fimmu.2022.878244. 

Surve MV, Bhutda S, Datey A, Anil A, Rawat S, Pushpakaran A, Singh D, Kim KS, Chakravortty D, Banerjee A. Heterogeneity in pneumolysin expression governs the fate of Streptococcus pneumoniae during blood-brain barrier trafficking. PLoS Pathog. 2018 Jul 16;14(7):e1007168. doi: 10.1371/journal.ppat.1007168. 

2. Serotype 1 and 8 with mild infections is not overall correct; see newer publications: doi: 10.1038/s41467-020-15751-6. doi: 10.1016/j.tim.2021.11.007.

We agree, the text has been modified to include the suggested publications: 

“It is worth noting that serotypes 1 and 8 have been shown to harbor mutations in the ply gene that annul this main characteristic and cause a much milder disease, due to non-hemolytic allele (sequence type, ST306) allows adaptation to an intracellular lifestyle (Jefferies et al., 2007; Badgujar et al., 2020). However, in sub-Saharan Africa, serotype 1 causes invasive pneumococcal disease due to an increased production of autolysin and hemolytic pneumolysin alleles (ST217) (Jacques et al., 2020). This serotype is a major cause of invasive pneumococcal disease globally, especially in Africa, South America, and Asia, with geographically distinct STs which form three genetic clusters designated as lineage A, B, and C (Chaguza et al., 2022). (lines 117-125).

Jefferies, J. M. C.; Johnston, C. H. G.; Kirkham, L.-A. S.; Cowan, G. J. M.; Ross, K. S.; Smith, A.; Clarke, S. C.; Brueggemann, A. B.; George, R. C.; Pichon, B. Presence of Nonhemolytic Pneumolysin in Serotypes of Streptococcus Pneumoniae Associated with Disease Outbreaks. Journal of Infectious Diseases 2007, 196 (6), 936–944. https://doi.org/10.1086/520091.

Badgujar DC, Anil A, Green AE, Surve MV, Madhavan S, Beckett A, Prior IA, Godsora BK, Patil SB, More PK, Sarkar SG, Mitchell A, Banerjee R, Phale PS, Mitchell TJ, Neill DR, Bhaumik P, Banerjee A. Structural insights into loss of function of a pore forming toxin and its role in pneumococcal adaptation to an intracellular lifestyle. PLoS Pathog. 2020 Nov 20;16(11):e1009016. doi: 10.1371/journal.ppat.1009016. 

Jacques LC, Panagiotou S, Baltazar M, Senghore M, Khandaker S, Xu R, Bricio-Moreno L, Yang M, Dowson CG, Everett DB, Neill DR, Kadioglu A. Increased pathogenicity of pneumococcal serotype 1 is driven by rapid autolysis and release of pneumolysin. Nat Commun. 2020 Apr 20;11(1):1892. doi: 10.1038/s41467-020-15751-6. 

Chaguza C, Yang M, Jacques LC, Bentley SD, Kadioglu A. Serotype 1 pneumococcus: epidemiology, genomics, and disease mechanisms. Trends Microbiol. 2022 Jun;30(6):581-592. doi: 10.1016/j.tim.2021.11.007. 

3. It is not correct that oligomerization is with up to 400 and 500 toxin monomers (the pore is a 400 Å); these are approx. 42 monomers – see publications (e.g. doi: 10.7554/eLife.23644)

We stand corrected, the text has been modified as follows: “The PLY pore is a 400 Å ring of 42 membrane-inserted monomers (Lawrence et al., 2015; Marshall et al., 2015; van Pee et al., 2017)”. (line 155)

van Pee, K.; Neuhaus, A.; D’Imprima, E.; Mills, D. J.; Kühlbrandt, W.; Yildiz, Ö. CryoEM Structures of Membrane Pore and Prepore Complex Reveal Cytolytic Mechanism of Pneumolysin. Elife 2017, 6. https://doi.org/10.7554/eLife.23644.

Marshall, J. E.; Faraj, B. H. A.; Gingras, A. R.; Lonnen, R.; Sheikh, M. A.; El-Mezgueldi, M.; Moody, P. C. E.; Andrew, P. W.; Wallis, R. The Crystal Structure of Pneumolysin at 2.0 Å Resolution Reveals the Molecular Packing of the Pre-Pore Complex. Sci Rep 2015, 5, 13293. https://doi.org/10.1038/srep13293.

Lawrence, S. L.; Feil, S. C.; Morton, C. J.; Farrand, A. J.; Mulhern, T. D.; Gorman, M. A.; Wade, K. R.; Tweten, R. K.; Parker, M. W. Crystal Structure of Streptococcus Pneumoniae Pneumolysin Provides Key Insights into Early Steps of Pore Formation. Sci Rep 2015, 5, 14352. https://doi.org/10.1038/srep14352.

4. Line 121: influx of cytosolic calcium?

Thanks for the comment. We have changed “cytosol calcium” for “mitochondrial calcium” (line 137)

5. Line 125: sensing TLR4 is under debate, this has to be mentioned because of contradictory results

The controversial interaction between PLY and TLR4 has been tackled as follows:

“The interaction between PLY and TLR4 remains controversial. Some authors showed that the interaction of PLY with TLR4 is involved in induce cytokine production and apoptosis (Malley et al., 2003; Srivastava et al., 2005), while other research showed that PLY activates the NLRP3/ACS inflammasome to enhance the secretion of pro-inflammatory cytokines IL-1β and IL-18 from macrophages and dendritic cells and contributes to the protection of the host from pneumococcal infection independent of TLR-4 and mediated by K+ influx (McNeela et al., 2010; Karmakar et al., 2015).” (lines 142-148).

Malley R, Henneke P, Morse SC, Cieslewicz MJ, Lipsitch M, Thompson CM, Kurt-Jones E, Paton JC, Wessels MR, Golenbock DT. Recognition of pneumolysin by Toll-like receptor 4 confers resistance to pneumococcal infection. Proc Natl Acad Sci U S A. 2003 Feb 18;100(4):1966-71. doi: 10.1073/pnas.0435928100. 

Srivastava A, Henneke P, Visintin A, Morse SC, Martin V, Watkins C, Paton JC, Wessels MR, Golenbock DT, Malley R. The apoptotic response to pneumolysin is Toll-like receptor 4 dependent and protects against pneumococcal disease. Infect Immun. 2005 Oct;73(10):6479-87. doi: 10.1128/IAI.73.10.6479-6487.2005.

Karmakar, M.; Katsnelson, M.; Malak, H. A.; Greene, N. G.; Howell, S. J.; Hise, A. G.; Camilli, A.; Kadioglu, A.; Dubyak, G. R.; Pearlman, E. Neutrophil IL-1β Processing Induced by Pneumolysin Is Mediated by the NLRP3/ASC Inflammasome and Caspase-1 Activation and Is Dependent on K+ Efflux. J Immunol 2015, 194 (4), 1763–1775. https://doi.org/10.4049/jimmunol.1401624.

McNeela, E. A.; Burke, A.; Neill, D. R.; Baxter, C.; Fernandes, V. E.; Ferreira, D.; Smeaton, S.; El-Rachkidy, R.; McLoughlin, R. M.; Mori, A.; Moran, B.; Fitzgerald, K. A.; Tschopp, J.; Pétrilli, V.; Andrew, P. W.; Kadioglu, A.; Lavelle, E. C. Pneumolysin Activates the NLRP3 Inflammasome and Promotes Proinflammatory Cytokines Independently of TLR4. PLoS Pathog 2010, 6 (11), e1001191. https://doi.org/10.1371/journal.ppat.1001191.

6. Line 159: the authors are not familiar with the newest publications; the statement is not correct anymore (see doi: 10.1182/bloodadvances.2020002372)

We have changed the text and added the reference: “Regarding prothrombotic effects, this has recently been shown not to be true. PLY does not activate platelets to form thrombus, rather it destroys them by forming pores in their membrane (Jahn et al., 2020; 2022) and destroy procoagulant microvesicles impaired coagulation of blood (Oehmcke-Hecht et al, 2022)” (lines 178-181).

Jahn K, Handtke S, Palankar R, Weißmüller S, Nouailles G, Kohler TP, Wesche J, Rohde M, Heinz C, Aschenbrenner AF, Wolff M, Schüttrumpf J, Witzenrath M, Hammerschmidt S, Greinacher A. Pneumolysin induces platelet destruction, not platelet activation, which can be prevented by immunoglobulin preparations in vitro. Blood Adv. 2020 Dec 22;4(24):6315-6326. doi: 10.1182/bloodadvances.2020002372. PMID: 33351126; PMCID: PMC7756997.

Jahn K, Kohler TP, Swiatek LS, Wiebe S, Hammerschmidt S. Platelets, Bacterial Adhesins and the Pneumococcus. Cells. 2022 Mar 25;11(7):1121. doi: 10.3390/cells11071121. 

Oehmcke-Hecht S, Maletzki C, Surabhi S, Siemens N, Khaimov V, John LM, Peter SM, Hammerschmidt S, Kreikemeyer B. Procoagulant Activity of Blood and Microvesicles Is Disturbed by Pneumococcal Pneumolysin, Which Interacts with Coagulation Factors. J Innate Immun. 2022 Jul 15:1-17. doi: 10.1159/000525479. 

7. Combining results and discussion will be highly beneficial for this Review.

We concur, and therefore Results and Discussion have been combined. 

8. When the effect of antibodies are discussed, the authors have also to consider these studies: doi: 10.1182/bloodadvances.2020002372, doi: 10.1055/a-1723-1880

Thanks for the comment. We have added the text and suggested references added: “In fact, the effect of PLY on platelets and erythrocytes can be neutralized by polyvalent human IgG and trimodulin (Wiebe et al., 2022). In a clinical trial in patients with S. pneumoniae CAP, treatment with trimodulin (182.6 mg/kg, for 5 consecutive days) improved survival relative to patients treated with placebo (Jahn et al., 2020). Drugs developed against toxins from other microorganisms, including monoclonal antibodies, antibody fragments, antibody mimetics, have recently been reviewed (Sakari et al., 2022). It is important to highlight the phase I clinical trial using liposomes to capture S. pneumoniae toxins (Laterre et al., 2019). (lines 634-640)

Wiebe F, Handtke S, Wesche J, Schnarre A, Palankar R, Wolff M, Jahn K, Voß F, Weißmüller S, Schüttrumpf J, Greinacher A, Hammerschmidt S. Polyvalent Immunoglobulin Preparations Inhibit Pneumolysin-Induced Platelet Destruction. Thromb Haemost. 2022 Jul;122(7):1147-1158. doi: 10.1055/a-1723-1880. 

Jahn K, Handtke S, Palankar R, Weißmüller S, Nouailles G, Kohler TP, Wesche J, Rohde M, Heinz C, Aschenbrenner AF, Wolff M, Schüttrumpf J, Witzenrath M, Hammerschmidt S, Greinacher A. Pneumolysin induces platelet destruction, not platelet activation, which can be prevented by immunoglobulin preparations in vitro. Blood Adv. 2020 Dec 22;4(24):6315-6326. doi: 10.1182/bloodadvances.2020002372. 

Sakari M, Laisi A, Pulliainen AT. Exotoxin-Targeted Drug Modalities as Antibiotic Alternatives. ACS Infect Dis. 2022 Mar 11;8(3):433-456. doi: 10.1021/acsinfecdis.1c00296. 

Laterre PF, Colin G, Dequin PF, Dugernier T, Boulain T, Azeredo da Silveira S, Lajaunias F, Perez A, François B. CAL02, a novel antitoxin liposomal agent, in severe pneumococcal pneumonia: a first-in-human, double-blind, placebo-controlled, randomised trial. Lancet Infect Dis. 2019 Jun;19(6):620-630. doi: 10.1016/S1473-3099(18)30805-3. 

Minor comments:

1. Line 94: change LPXTG proteins to “sortase-anchored proteins”

Thanks for the comment. We have changed LPXTG proteins to “sortase-anchored proteins” (line 95)

2. Line 99: Re-check your statement in line 99-100 that pneumolysin is expressed in the late-log phase. This is not correct, instead PLY is produced constitutively and the free or released PLY is higher in the late log phase because of autolysis and the heterogeneity in the culture.

The paragraph has been modified as follows: “PLY is produced constitutively, but free toxin is higher in the late log phase due to the presence of a defined threshold concentration of extracellular autolysin (LytA) which dictates the onset of autolysis. The entry into the stationary phase due to nutrient depletion sensitizes cells to the effect of LytA, while during exponential growth they are protected from the action of this enzyme (Mellroth et al., 2012; Flores-Kim et al., 2022). Other investigations have also revealed the release of PLY in the extracellular vesicles (Codemo et al., 2018). On the other hand, a phenotypic heterogenicity has been demonstrated in terms of the level of expression of PLY, which helps the dispersion of the pneumococcus through the host (Surve et al., 2018; 2019).” (lines 103-110).

Mellroth, P.; Daniels, R.; Eberhardt, A.; Rönnlund, D.; Blom, H.; Widengren, J.; Normark, S.; Henriques-Normark, B. LytA, Major Autolysin of Streptococcus Pneumoniae, Requires Access to Nascent Peptidoglycan. J Biol Chem 2012, 287 (14), 11018–11029. https://doi.org/10.1074/jbc.M111.318584.

Flores-Kim, J.; Dobihal, G. S.; Bernhardt, T. G.; Rudner, D. Z. WhyD Tailors Surface Polymers to Prevent Premature Bacteriolysis and Direct Cell Elongation in Streptococcus Pneumoniae. Elife 2022, 11, e76392. https://doi.org/10.7554/eLife.76392.

Codemo M, Muschiol S, Iovino F, Nannapaneni P, Plant L, Wai SN, Henriques-Normark B. Immunomodulatory Effects of Pneumococcal Extracellular Vesicles on Cellular and Humoral Host Defenses. mBio. 2018 Apr 10;9(2):e00559-18. doi: 10.1128/mBio.00559-18. 

Surve, M. V.; Banerjee, A. Cell-to-Cell Phenotypic Heterogeneity in Pneumococcal Pathogenesis. Future Microbiol 2019, 14, 647–651. https://doi.org/10.2217/fmb-2019-0096.

Surve, M. V.; Bhutda, S.; Datey, A.; Anil, A.; Rawat, S.; Pushpakaran, A.; Singh, D.; Kim, K. S.; Chakravortty, D.; Banerjee, A. Heterogeneity in Pneumolysin Expression Governs the Fate of Streptococcus Pneumoniae during Blood-Brain Barrier Trafficking. PLoS Pathog 2018, 14 (7), e1007168. https://doi.org/10.1371/journal.ppat.1007168.

3. Line 103: again, this is not exact what happens; re-write after checking the literature

The text has been re-written as follows: “PLY does not have attachment motifs, however the toxin localizes to the cell envelope of actively growing cells, where its release and activity is controlled by the composition of the peptidoglycan, specifically by the proportion of branched stem peptides that vary throughout the cell cycle and between different strains (Greene et al., 2015)” (lines 110-114)

Greene, N. G.; Narciso, A. R.; Filipe, S. R.; Camilli, A. Peptidoglycan Branched Stem Peptides Contribute to Streptococcus Pneumoniae Virulence by Inhibiting Pneumolysin Release. PLoS Pathog 2015, 11 (6), e1004996. https://doi.org/10.1371/journal.ppat.1004996

---

## [Decision Letter · Decision Letter 1]

28 Feb 2023

Pneumolysin as a target for new therapies against pneumococcal infections: A systematic review

PONE-D-22-26630R1

Dear Dr. DEL MAR,

We’re pleased to inform you that your manuscript has been judged scientifically suitable for publication and will be formally accepted for publication once it meets all outstanding technical requirements.

Kind regards,

Raj Kumar Koiri

Academic Editor

PLOS ONE

Additional Editor Comments (optional):

The author has suitably addressed the comments raised by the reviewers. The manuscript can be accepted.

Reviewers' comments:

Reviewer's Responses to Questions

**Comments to the Author**

1. If the authors have adequately addressed your comments raised in a previous round of review and you feel that this manuscript is now acceptable for publication, you may indicate that here to bypass the “Comments to the Author” section, enter your conflict of interest statement in the “Confidential to Editor” section, and submit your "Accept" recommendation.

Reviewer #1: All comments have been addressed

Reviewer #3: All comments have been addressed

2. Is the manuscript technically sound, and do the data support the conclusions?

Reviewer #1: Yes

Reviewer #3: Yes

3. Has the statistical analysis been performed appropriately and rigorously? 

Reviewer #1: Yes

Reviewer #3: N/A

4. Have the authors made all data underlying the findings in their manuscript fully available?

Reviewer #1: Yes

Reviewer #3: Yes

5. Is the manuscript presented in an intelligible fashion and written in standard English?

Reviewer #1: Yes

Reviewer #3: Yes

6. Review Comments to the Author

Reviewer #1: (No Response)

Reviewer #3: The authors have adequately modified their manuscript. The auhtors have considered all the comments and suggestions rased by reviewers. I have no further comments to the manuscript and suggest to accept the manuscript in the current revised version.

7. PLOS authors have the option to publish the peer review history of their article (what does this mean?). If published, this will include your full peer review and any attached files.

Reviewer #1: **Yes: **Dr. Saikat Banerjee MD

Reviewer #3: **Yes: **Sven Hammerschmidt

---

## [Editor Report · Acceptance letter]

13 Mar 2023

PONE-D-22-26630R1 

Pneumolysin as a target for new therapies against pneumococcal infections: A systematic review 

Dear Dr. Del Mar García-Suárez:

I'm pleased to inform you that your manuscript has been deemed suitable for publication in PLOS ONE. Congratulations! Your manuscript is now with our production department. 

Kind regards, 

on behalf of

Dr. Raj Kumar Koiri 

Academic Editor

PLOS ONE